



# Jarzynski equality and Crooks relation for local models of air-sea interaction

Achim Wirth[1] and Florian Lemarié:[2]

[1]Univ. Grenoble Alpes, CNRS, Grenoble INP, LEGI, 38000 Grenoble, France
[2]Univ. Grenoble Alpes, Inria, CNRS, Grenoble INP, LJK, 38000 Grenoble,France

**Correspondence:** achim.wirth@legi.cnrs.fr

**Abstract.** We show using a hierarchy of local models of air-sea interaction that the most prominent of the work theorems, the Jarzynski equality and the Crooks relation can be applied to air-sea interaction. In the more idealized models, with and without a Coriolis force, the variability is provided from a Gaussian white-noise which modifies the shear between the atmosphere and the ocean. The dynamics is Gaussian and the Jarzynski equality and Crooks relation can be obtained analytically solving

stochastic differential equations. The more involved model consists of interacting atmospheric and oceanic boundary-layers, where only the dependence on the vertical direction is resolved, the turbulence is modeled through standard turbulent models and the stochasticity comes from a randomized drag coefficient. It is integrated numerically and can give rise to a non-Gaussian dynamics. Also in this case the Jarzynski equality allows for calculating a dynamic-beta $\beta_D$ of the turbulent fluctuations (the equivalent of the thermodynamic-beta $\beta = (k_B T)^{-1}$ in thermal fluctuations). The Crooks relation gives the $\beta_D$ as a function of

the magnitude of the work fluctuations. It is well defined (constant) in the Gaussian models and can show a slight variation in the more involved models. This demonstrates that recent concepts of stochastic thermodynamics used to study micro-systems subject to thermal fluctuations can further the understanding of geophysical fluid dynamics with turbulent fluctuations.

## 1  Introduction

To better understand the interactions between different components of the climate system is an important and difficult task. The problem lies in the different science proper to each component leading to disparate processes, evolving on dissimilar scales in space and time. This heterogenity complexifies the research, from an observational, theoretical and numerical perspective. Air-sea interaction is one example. The exchange of heat, momentum and matter between the atmosphere and the ocean has a strong influence on our climate (Stocker et al., 2013). In the present work only the exchange of momentum is considered.

It is caused by the shear at the sea surface due to the difference between the atmospheric winds and the ocean currents in the corresponding planetary boundary layers. For a general discussion on air-sea interaction we refer to Csanady (2001). The atmospheric winds are usually faster than the ocean currents and therefore the atmosphere mostly looses energy at the interface





by friction and the ocean mostly gains energy (e.g. Wirth, 2019). The energy exchange is not conservative and most of the energy is dissipated (Duhaut and Straub, 2006; Wirth, 2018).

Since the work of Einstein (1906) (see also Einstein, 1956; Perrin, 2014), fluctuations are the focus of research in statistical mechanics, which was traditionally concerned with averages. Fluctuations in a thermodynamic system usually appear at spatial scales which are small enough so that thermal, molecular, motion leaves an imprint on the dynamics, as was first noted by Einstein (1906) (see also Einstein, 1956; Perrin, 2014). The importance of fluctuations is, however, not restricted to small systems where thermal fluctuations are important, since they leave their imprint on the dynamics at all scales when (not necessarily thermal) fluctuations are strong enough. A typical example, of non-thermal fluctuations, is fluctuating turbulent fluid motion (e.g. Frisch, 1995). The average motion of a turbulent fluid can not be understood without some knowledge about the turbulent fluctuations. The importance of turbulent fluctuations is especially pronounced in geophysical flows, which are highly anisotropic due to the influence of gravity. This leads to a quasi two-dimensional dynamics and an energy cascade from small to large scales and strong fluctuations (see Boffetta and Ecke, 2012, for a recent review on 2D turbulence). Likewise, the air-sea interaction on hourly to climatic time scales can not be understood without some knowledge of the fluctuations at smaller and faster scales (see McWilliams and Huckle, 2006; Shrira et al., 2020). Furthermore, in many natural systems the focus is on the fluctuations rather than on an average state. Examples are weather and climate dynamics, where we focus on the fluctuations of the same system on different time scales. For the weather the time scale of interest is from roughly an hour to a week, for the climate the focus is from tenths to thousands of years. As processes with very different timescales intervene, the system is not in a stationary state at those time scales, but is constantly evolving in time. The different components of the system exchange energy, they do work on each other. The exchange of energy between fluctuating components is the subject of the present work.

A recent concept, which is presently subject of attention when non-equilibrium thermal systems are considered, are work theorems. The most prominent ones are the Jarzynski equality (Jarzynski, 1997) and the Crooks relation (Crooks, 1998). Rather than looking at average values of the thermodynamic variables they consider their probability density functions (pdf) which allow to replace inequalities of equilibrium statistical mechanics by equalities. As an example: the second law of thermodynamics states that the work $W$ performed on a system is larger or equal than the increase $\Delta G$ in its free energy: $W \geq \Delta G$. When the work is seen as a fluctuating quantity $w$, which differs even when a specific process is repeated with the same deterministic forcing protocol, but subject to thermal fluctuations, the Jarzynski equality says that $\langle \exp(-\beta w) \rangle = \exp(-\beta \Delta G)$, where the average $\langle \rangle$ is taken over the ensemble of thermal fluctuations. This not only includes the second law on average, but also says that individual exceptions have to exist (see section 2). When thermal fluctuations are considered, the (thermodynamic-) $\beta = (k_B T)^{-1}$ is the inverse of the product of the Boltzmann constant and the temperature. In the case of air-sea interaction, considered here, the dynamic-$\beta$, (denoted $\beta_D$) is the inverse of an energy related to the macroscopic turbulent fluctuations and does not have the meaning of an inverse temperature.

The here discussed work theorems are different but related to fluctuation theorems considered in Wirth (2018) and Wirth (2019). In a recent review Seifert (2012) presents the relation of Fluctuation Theorems, the Jarzynski equality, the Crooks relation and other recent concepts of non-equilibrium thermodynamics and develops a unifying framework. Work theorems are





considered based on different approaches, Hamiltonian dynamics subject to an external forcing, Foker-Planck equations and Langevin dynamics (see Seifert, 2012, for a review). Here only the last approach is used.

The concepts developed for micro dynamics with fluctuations due to thermal motion are here applied to macroscopic fluid dynamics, where an atmospheric planetary boundary layer interacts with an oceanic mixed layer. In this case the fluctuations

are due to the smaller-scale turbulence in both layers. The concepts of fluctuation theorems have been previously applied to cases with turbulent rather than thermal fluctuations. Examples are the experimental data of the drag-force exerted by a turbulent flow (Ciliberto et al., 2004) and the local entropy production in Rayleigh-Bénard convection (Shang et al., 2005).

A system that is subject to an external forcing typically evolves in time, it is in a non-stationary state. If there is a balance between external forcings and/or internal dissipation in such a way that ensemble averages do not evolve in time the system is

in a non-equilibrium stationary state. In the here considered work theorems a dissipative system is subject to forcing and also the average large scale quantities evolve in time, the dynamics is in a non-stationary non-equilibrium state.

The concepts of non-equilibrium statistical mechanics have been applied to momentum transfer between the atmosphere and the ocean in a non-rotating frame in Wirth (2018) and Wirth (2019). This was done by adapting the mathematics developed to study the movement of a Brownian particle. The present work prolongs this research by considering work relations and

extending it to the dynamics in a rotating frame. The motion of a particle in a rotating frame is similar to Brownian motion of a charged particle in a magnetic field, a problem which is studied since Taylor (1961) (see also Czopnik and Garbaczewski, 2001). The structure of the equations is identical, when the Larmor frequency of a charged particle in a magnetic field is replaced by the Coriolis frequency. The passage from a non-rotating frame to a rotating frame is, however, far from straight forward, for principally two reasons. First, the dynamics is no-longer invariant by time-reversal, even in the non-dissipative

limit. In the words of statistical mechanics: detailed balance, which is the basis of many analytical results, is lost. Secondly, it is not clear that results from simple models that do not explicitly resolve the vertical structure in the atmospheric and oceanic boundary layer are useful to investigate the situation in a rotating frame with a Coriolis force (see McWilliams and Huckle, 2006). Indeed, the dynamics in the planetary boundary-layer shows a strong dependence with the vertical coordinate, not only in magnitude, but also in direction as determined by Ekman (1905). Analytic solutions for time evolution are only available in

special cases (see Shrira et al., 2020). In the present pedagogical approach to the subject we therefore work with a hierarchy of three models. The first model is a linear zero-dimensional one-component model (1D velocity vector). We analytically prove the validity of the work theorems by solving the corresponding stochastic differential equation (SDE). In the second model the Coriolis force is added and it has two horizontal components (2D velocity vector). The work theorems are again proven analytically. The third model is a fully non-linear model, explicitly resolving the vertical dependence of the interacting

atmospheric and oceanic boundary-layer, which is integrated numerically.

In the next section we introduce the theory of stochastic thermodynamics and work relations applied to air-sea interaction. The models are introduced and solved, using stochastic calculus, in section 3. As the concepts are new to the field (see Ghil, 2019, for a historical perspective) we present all the calculations in detail for pedagogical purposes and also to show that most of it reduces to linear algebra. We refer the reader not familiar with stochastic differential equations to Dijkstra (2013);





Franzke et al. (2015). The results, for the three models of our model hierarchy, are discussed in section 4 and we end with some conclusions in section 5.

## 2 Theory

### 2.1 Model

#### 5  2.1.1  The 1D two components model (1D2C)

We consider the turbulent momentum transfer between the atmospheric and the oceanic planetary boundary-layer, which are coupled by a frictional force. The atmospheric layer is also subject to a deterministic forcing imposed from the exterior through a pressure gradient. The dynamics in the boundary layers is investigated using a Reynolds decomposition, in which the fast fluctuations of the three-dimensional velocity are separated from the slowly evolving component of the horizontal velocity field (called "velocity field" in the sequel). The horizontal variations of the velocity field are neglected. This is justified in a

10 local model by the disparity of the vertical and horizontal scales. The atmospheric planetary boundary layer is a few hundreds of meters thick. The oceanic planetary boundary layer spans a few tenths of meters in the vertical. The velocity field in both layers varies considerably over the thickness of the corresponding boundary layer. Horizontal variations are imposed by the weather systems that forces the dynamics and typically extends 1000km in the horizontal. This leads to a classical model of

15 the planetary boundary layers (introduced by Ekman, 1905), which depends on the vertical direction (1D) and resolves the two horizontal components (2C) of the velocity vector $\widetilde{\mathbf{u}}_a(z,t) = (\widetilde{u}_a(z,t), \widetilde{v}_a(z,t))$ [1], this 1D2C model is given by an evolution equation of both velocity components:

$$
\begin{cases}
\partial_t \widetilde{u}_a(z,t) = \phantom{-} f\widetilde{v}_a(z,t) + \partial_z[\nu_a(z,t)\partial_z\widetilde{u}_a(z,t)] + \widetilde{F}_x(t) & \text{(1a)} \\
\partial_t \widetilde{v}_a(z,t) = -f\widetilde{u}_a(z,t) + \partial_z[\nu_a(z,t)\partial_z\widetilde{v}_a(z,t)] + \widetilde{F}_y(t), & \text{(1b)}
\end{cases}
$$

where $f$ is the Coriolis frequency, $\nu_a(z,t)$ the turbulent viscosity, and $\widetilde{\mathbf{F}} = (\widetilde{F}_x, \widetilde{F}_y)$ a forcing provided by a large-scale pressure

gradient, which is independent of the vertical direction. The turbulent viscosity $\nu_\alpha(z,t)$ parameterizes the effect of the not explicitly resolved fluctuations on the velocity field, it is calculated through a turbulent closure scheme. The atmosphere extends over $z \in [0, h_a]$ and the boundary conditions are (Neumann at top and bottom):

$$
(\partial_z \widetilde{\mathbf{u}}_a)_{z=h_a} = 0 \tag{2a}
$$

$$
(\nu_a \partial_z \widetilde{\mathbf{u}}_a)_{z=0} = \frac{\boldsymbol{\tau}}{\rho_a} \tag{2b}
$$

---

[1]The superscript ~ is used to characterize a variable which is function of $z$ and $t$





where $\rho_a$ is a constant atmospheric density. The ocean is also governed by model (1) where all subscripts are changed ($_a \to _o$), the forcing vanishes, the domain extends over $z \in [-h_o, 0]$, and the boundary conditions are (Neumann at bottom and top):

$$(\partial_z \tilde{\mathbf{u}}_o)_{z=-h_o} = 0 \tag{3a}$$

$$(\nu_o \partial_z \tilde{\mathbf{u}}_o)_{z=0} = \frac{\tau}{\rho_o}. \tag{3b}$$

The surface friction $\tau = (\tau_y, \tau_y)$ is parameterized as a function of the velocity difference between the atmospheric and oceanic velocity near the interface $z = 0$. Either a linear Rayleigh friction (i.e. parameterized to be linearly proportional to the relative wind) is employed:

$$\tau = \rho_o h_o S(\tilde{\mathbf{u}}_a(\delta_a) - \tilde{\mathbf{u}}_o(-\delta_o)), \tag{4}$$

with $S^{-1}$ an oceanic friction time, or a quadratic drag law:

$$\tau = \rho_o c_d |\tilde{\mathbf{u}}_a(\delta_a) - \tilde{\mathbf{u}}_o(-\delta_o)|(\tilde{\mathbf{u}}_a(\delta_a) - \tilde{\mathbf{u}}_o(-\delta_o)), \tag{5}$$

with $\delta_a \ll h_a$ and $\delta_o \ll h_o$. Here the drag coefficient $c_d$ is defined relative to the ocean, the equivalent drag coefficient for the atmosphere is obtained by multiplying $c_d$ by $\rho_o/\rho_a$.

### 2.1.2 The 0D two components model (0D2C)

The 0D version of the 1D2C model (1) is obtained by integration over the vertical extent of the corresponding layer normalized
by the layer thickness. Introducing

$$\mathbf{u}_a = (u_a, v_a) = \frac{1}{h_a} \int_0^{h_a} \tilde{\mathbf{u}}_a(z) dz \qquad \mathbf{u}_o = (u_o, v_o) = \frac{1}{h_o} \int_{-h_o}^{0} \tilde{\mathbf{u}}_o(z) dz, \tag{6}$$

and using the boundary conditions (eqs. (2) and (3)) as well as the linear Rayleigh friction (4), we obtain the following 0D2C model for the atmosphere:

$$\begin{cases} \partial_t u_a = \phantom{-}fv_a - mS(u_a - u_o) + F_x & \text{(7a)} \\ \partial_t v_a = -fu_a - mS(v_a - v_o) + F_y & \text{(7b)} \end{cases}$$

where $m = \dfrac{\rho_o h_o}{\rho_a h_a}$ is the mass ratio between the oceanic layer and the atmospheric layer, and $\mathbf{F}(t) = (F_x, F_y)$ is analogous to $\widetilde{\mathbf{F}}(t)$. Similarly for the ocean, we have

$$\begin{cases} \partial_t u_o = \phantom{-}fv_o + S(u_a - u_o) & \text{(8a)} \\ \partial_t v_o = -fu_o + S(v_a - v_o), & \text{(8b)} \end{cases}$$





The momentum exchange between the layers due to unresolved turbulent motion is parameterized by a random force $\boldsymbol{\zeta} = (\zeta_x, \zeta_y)$, added to the friction law. The model under consideration in the following thus reads

$$\begin{cases} \partial_t u_a = & f v_a - mS(u_a - u_o) + F_x + \dfrac{m}{M}\zeta_x & \text{(9a)} \\[2mm] \partial_t v_a = & -f u_a - mS(v_a - v_o) + F_y + \dfrac{m}{M}\zeta_y & \text{(9b)} \end{cases}$$

$$\begin{cases} \partial_t u_o = & f v_o + S(u_a - u_o) - \dfrac{1}{M}\zeta_x & \text{(9c)} \\[2mm] \partial_t v_o = & -f u_o + S(v_a - v_o) - \dfrac{1}{M}\zeta_y, & \text{(9d)} \end{cases}$$

where the stochastic noise has been scaled by $M = m + 1$ to simplify the algebra in the following. Note that $M$ is the total mass per unit surface. The analytical solution to the coupled model (9) is given in appendix B. When the Coriolis parameter vanishes ($f = 0$) and the linear friction law is used, the dynamics in the two horizontal directions is uncoupled. In this case a simple 0D1C model can be obtained by setting $v_a = 0$ in (9a) (resp. $v_o = 0$ in (9c)) and discarding (9b) (resp. (9d)).

Let us also introduce the integrated mode, which gives the momentum integrated over $[-h_o, h_a]$, and the shear mode:

$$\mathbf{u}_I = \mathbf{u}_a + m\mathbf{u}_o \tag{10a}$$

$$\mathbf{u}_S = \mathbf{u}_a - \mathbf{u}_o. \tag{10b}$$

The shear and the turbulence in the atmosphere and the ocean do not affect the integrated momentum $\mathbf{u}_I$. The two layers only interact by friction, which acts as a damping on the shear mode (see A1 and B1). The only remaining two parameters in the problem are the constant mass ratio of the oceanic versus the atmospheric layer, $m$, and the function $S$. For the case of linear Rayleigh friction (4), $S$ is constant. When the turbulent, quadratic, friction-law (5) applies we have $S = c_d |\mathbf{u_S}|/h_o$, with a constant drag coefficient $c_D$.

The departures from the vertical average in the atmosphere and the ocean are given by:

$$\mathbf{u}'_a(z) = \tilde{\mathbf{u}}_a(z) - \mathbf{u}_a \tag{11a}$$

$$\mathbf{u}'_o(z) = \tilde{\mathbf{u}}_o(z) - \mathbf{u}_o. \tag{11b}$$

The interaction between the different components is schematized in fig. 1. The dynamics of the integrated mode, $\mathbf{u}_I$, does not depend on the shear $\boldsymbol{\tau}$, as can be verified when (10a) is combined with (9) (see A1and B1). Newton's laws insure that the dynamics of the integrated mode is independent of the interior dynamics, that is from $\mathbf{u}_S$, $\mathbf{u}'_a$, $\mathbf{u}'_o$, $\nu_a(z)$ and $\nu_o(z)$. This property is lost when a dependence on the horizontal directions is included. Due to the boundary conditions it is also not subject to dissipation and therefore conserves its (kinetic) energy. The dynamics of the integrated mode is purely deterministic and the work $W_I$ done on it equals the increase of the free energy $\Delta G$. The shear mode $\mathbf{u}_S$ interacts with the internal modes in the atmosphere, $\mathbf{u}'_a$, and the ocean, $\mathbf{u}'_o$, through the shear at the interface $\boldsymbol{\tau}$. The dynamics in the shear mode does not depend explicitly on the internal viscosity $\nu_a(z), \nu_o(z)$ in the two layers, but only through $\mathbf{u}'_a$ and $\mathbf{u}'_o$. As $\mathbf{u}'_a$ has a vanishing





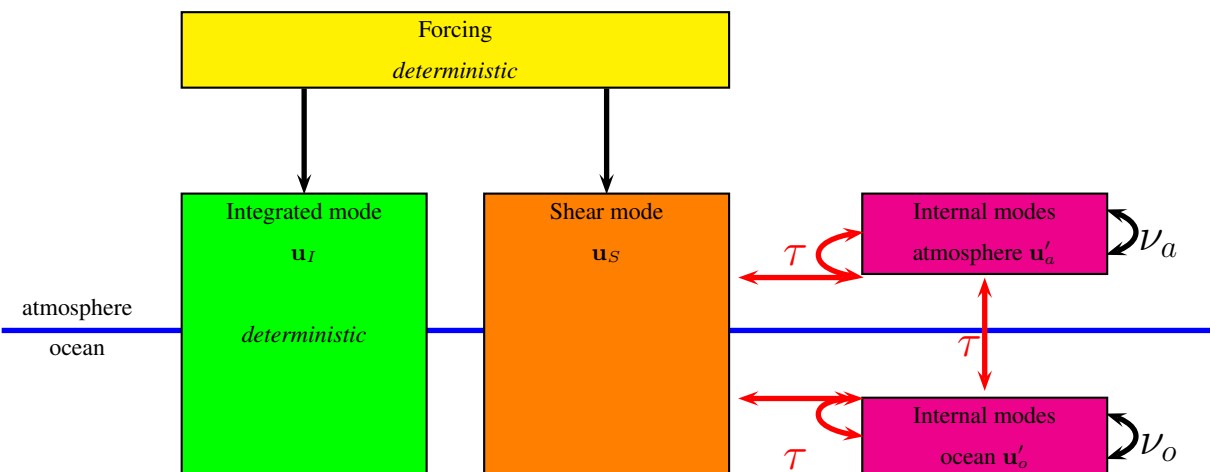

**Figure 1.** Schematic of the models considered: The integrated mode and the shear mode are forced. The integrated mode is decoupled from the rest of the dynamics. The shear mode is coupled to the internal modes of the atmosphere and the ocean by the surface stress. The internal modes in the atmosphere and the ocean depend on the eddy viscosity in each layer and the surface stress. the randomness arises thorough the surface friction $\tau$ (red color). In the 1D model the internal dynamics in the atmospheric and oceanic layer are explicitly resolved and a random noise is added to the surface friction coefficient. In the 0D model their influence on the shear mode is parameterized by a random noise.

vertical average, it is not forced by the pressure gradient. The dynamics of $\mathbf{u}'_a$ depends explicitly on $\mathbf{u}_S$, $\mathbf{u}'_a$, $\mathbf{u}'_o$ and $\nu_a(z)$. The same applies to the ocean. In the 0D models the effect of the internal modes (magenta boxes in fig. 1) on the shear mode is modeled by a stochastic noise. In the 1D model the internal dynamics is resolved explicitly, stochastic noise is added to the drag coefficient only and enters the dynamics through the shear at the interface (red arrows in fig. 1), see section 3.3 for more
5  details on this point. Looking at air-sea interaction in terms of modes is not only a technical simplification but also emphasizes the view of seeing the atmosphere and the ocean as a combined system rather than a separate atmospheric and oceanic layer that act on each other. From eqs. (10a) and (10b) we easily get that:

$$\mathbf{u_a} = \frac{1}{M}(\mathbf{u_I} + m\mathbf{u_S}) \tag{12a}$$

$$\mathbf{u_o} = \frac{1}{M}(\mathbf{u_I} - \mathbf{u_S}). \tag{12b}$$

10  with $M = m + 1$ the total mass per unit surface.

## 2.2 Stochastic Thermodynamics

The concept of stochastic thermodynamics was introduced by Sekimoto (1998) (see also Seifert, 2012). Rather than considering the classical dynamics described by Hamilton's equations over the entire phase-space of all microstates, on one side, or the





averaged thermodynamic quantities, without internal dynamics, on the other, it takes an intermediate position by looking at mesostates (also called statistical states). A mesostate does not completely determine the microstate of the system but represents an ensemble of microstates. It is therefore not described by a sharp value but a pdf. Its mathematical framework are the Langevin equation and the stochastic differential equations, which describe the evolution of a pdf. A dynamics with a deterministic and

stochastic part that interact. Such approach is adapted when external forces only constrain part of the dynamics as the internal response of the system is too involved (chaotic or turbulent), so that it can only be described in a stochastic sense. If a specific force is applied to a system the outcome depends on the initial microstate that is usually not precisely known and its evolution has a random component. By considering the evolution of the pdf, which takes into account the uncertainty in the microstates that influence on the system, we obtain a deterministic evolution of the pdf.

We here apply these concepts to air-sea interaction the "heat", the source of the fluctuation, in our approach is (small-scale) turbulent motion, all that is represented in magenta and red in fig. 1. The macroscopic variable are the slowly-varying vertically-averaged velocities $\mathbf{u}_a$, $\mathbf{u}_o$ or equivalently modes $\mathbf{u}_I$, $\mathbf{u}_S$. In analogy to the first law of thermodynamics we write:

$$dW = dV - dQ. \tag{13}$$

The work applied to the system by the external force $\mathbf{F} = (F_x, F_y)$ in (9) is:

$$dW \quad = \quad \mathbf{F} \cdot d\mathbf{x}_a = \mathbf{F} \cdot \mathbf{u}_a dt \tag{14}$$

For the sake of readability, in the sequel we will omit the dot symbol "·" in vector products. $V$, $dV$ and $dQ$ should be understood as scalar quantities in the following. The internal (kinetic) energy is:

$$V \quad = \quad \frac{1}{2}(\mathbf{u}_a^2 + m\mathbf{u}_o^2) = \frac{1}{2M}(\mathbf{u}_I^2 + m\mathbf{u}_S^2) \tag{15}$$

$$dV \quad = \quad \mathbf{u}_a d\mathbf{u}_a + m\mathbf{u}_o d\mathbf{u}_o = \frac{1}{M}(\mathbf{u}_I d\mathbf{u}_I + m\mathbf{u}_S d\mathbf{u}_S), \tag{16}$$

and the heat provided to the system ($Q < 0$ as friction dissipates heat):

$$dQ = dV - dW = \mathbf{u}_a d\mathbf{u}_a + m\mathbf{u}_o d\mathbf{u}_o - \mathbf{F}\mathbf{u}_a dt = \frac{1}{M}(\mathbf{u}_I d\mathbf{u}_I + m\mathbf{u}_S d\mathbf{u}_S) - \mathbf{F}\frac{1}{M}(\mathbf{u}_I + m\mathbf{u}_S)dt. \tag{17}$$

The dissipation to heat is given by:

$$\frac{dQ}{dt} \quad = \quad \mathbf{u}_a \frac{d\mathbf{u}_a}{dt} + m\mathbf{u}_o \frac{d\mathbf{u}_o}{dt} - \mathbf{F}\mathbf{u}_a = \mathbf{u}_a(-Sm\mathbf{u}_a + Sm\mathbf{u}_o + \mathbf{F}) + m\mathbf{u}_o(-S\mathbf{u}_o + S\mathbf{u}_a) - \mathbf{F}\mathbf{u}_a$$

$$= \quad -Sm(\mathbf{u}_a - \mathbf{u}_o)^2 = -Sm\mathbf{u}_S^2. \tag{18}$$

To derive the second line we used eqs. (9a - 9d). The shear force between the layers is $S\mathbf{u}_S$, when the friction law is linear $S$ is a constant otherwise it is a function of the shear. Furthermore if we consider a process that starts at $A$ and finishes at $B$ we have:

$$W(A \rightarrow B) \quad = \quad \int_{x(A)}^{x(B)} \mathbf{F} d\mathbf{x}_a = \int_{t(A)}^{t(B)} \mathbf{F}\mathbf{u}_a dt = \frac{1}{M}\int_{t(A)}^{t(B)} \mathbf{F}(\mathbf{u}_I + m\mathbf{u}_S)dt \tag{19}$$





The free energy is $\Delta G = V(\infty) - V(A) = \frac{1}{2M}\mathbf{u}_I^2$, the energy in the integrated mode, as the energy in the shear mode is dissipated away in time. Note that $\frac{dQ}{dt} < 0$ and therefore $\Delta G < W(A \to B)$ and $-\Delta G < W(B \to A)$, which leads to $-W(B \to A) < \Delta G < W(A \to B)$. This means that more work than the free energy has to be provided to go from A to B and less work is recuperated than the free energy on the reverse (conjugated) path. In a cyclic process $B = A$, $\Delta G = 0$ and all the work injected in the system is ultimately dissipated.

## 2.3 Forward, inverse and reverse Processes

The forcing protocol on the time interval $[0, T]$ is given by:

$$\tilde{\mathbf{F}}^{\mathbf{f}}_{A \to B}(t) \quad = \quad \mathbf{F}(t) \tag{20}$$

where the function $\mathbf{F}$ has a compact support within the interval $[0, T]$, It is important to note that even if the system evolves (relaxes) outside the interval $[0, T]$ no work is performed on the system as the force is vanishing. When the Coriolis force is present the system is generally not in a stationary state after the forcing, but performs inertial oscillations. To bring the system back to the initial state an inverse protocol has to be performed at precisely a multiple of the inertial period after the forward protocol:

$$\tilde{\mathbf{F}}^{\mathbf{i}}_{B \to A}(t) \quad = \quad -\mathbf{F}(t - \frac{2\pi}{f}n), \ \ n \in \mathbb{N}. \tag{21}$$

For a reverse protocol it is required that the forcing is:

$$\tilde{\mathbf{F}}^{\mathbf{r}}_{B \to A}(t) \quad = \quad -\mathbf{F}(T - t + t_0). \tag{22}$$

To satisfy both conditions we impose the symmetries:

$$\mathbf{F}(T/2 - t) = \mathbf{F}(T/2 + t) \ \ \text{and} \ \ t_0 = \frac{2\pi}{f}n, \ \ n \in \mathbb{N}. \tag{23}$$

If we neglect the turbulence in both layers, which is modeled by a stochastic term, the dynamics is deterministic. The forward process, starting from rest, is given by:

$$\mathbf{u}_I^f(0) \quad = \quad \mathbf{u}_S^f(0) = 0 \qquad \to \qquad \mathbf{u}_I^f(T), \ \ \mathbf{u}_S^f(\infty) = 0 \tag{24}$$
$$\Delta G^f(\infty) \quad = \quad \Delta G, \qquad W^f = \Delta G + Q^f, \qquad Q^f(0, \infty) = Q \tag{25}$$

The reverse process starts from the converged state, is forced by $\tilde{\mathbf{F}}^{\mathbf{r}}_{B \to A}(t)$ for a period $T$ and then relaxes to rest at $t = \infty$:

$$\mathbf{u}_I^r(0) \quad = \quad \mathbf{u}_I^f(T), \ \mathbf{u}_S^r(0) = 0 \qquad \to \qquad \mathbf{u}_I^r(T), \ \ \mathbf{u}_S^r(\infty) = 0 \tag{26}$$
$$\Delta G^r(\infty) \quad = \quad -\Delta G, \qquad W^r = -\Delta G + Q^r, \qquad Q^r(0, \infty) = Q \tag{27}$$

Note that $-W^r \le \Delta G \le W^f$ which is a statement of the second law of thermodynamics. When the process is reversible then the equalities apply. Furthermore we always have $2\Delta G = W^f - W^r$. So far the dynamics considered was deterministic.

The turbulent motion within the system is due to internal dynamics and is modeled by stochastic terms. When noise is added in the linear model it does not interfere with the deterministic dynamics but simply adds to it. Furthermore, the force





is deterministic, so that the randomness in the work provides solely from the fluctuations in $\mathbf{u}_a$. As randomness resides only in the shear-mode the fluctuations in the work $w' = m \int_0^T \mathbf{F} \mathbf{u}'_S dt$ come from fluctuations of the shear mode $\mathbf{u}'_S$. Note that the vertical average of $\mathbf{u}'_a$ vanishes, so that the internal modes do not contribute to the work. When the noise terms are Gaussian and the friction linear, the velocities are Gaussian variables and so is the work performed on the layers and modes. The average

of these variables are given by the deterministic part ($\langle w^f \rangle = W^f$, $\langle w^r \rangle = W^r$) and the variance $\sigma_W^2(T)$ is obtained through the variance of the shear mode. In the case with an internal turbulent dynamics, $-W^r \leq \Delta G^f(\infty) \leq W^f$ is true for averages only, individual trajectories can be exceptions.

In the Gaussian case the pdf's for the forward and reverse processes are:

$$\mathrm{pdf}^f(w) = \frac{1}{\sqrt{2\pi}\sigma_W} \exp(-\frac{(w - W^f)^2}{2\sigma_W^2}) \tag{28}$$

$$\mathrm{pdf}^r(w) = \frac{1}{\sqrt{2\pi}\sigma_W} \exp(-\frac{(w - W^r)^2}{2\sigma_W^2}) \tag{29}$$

Note that the pdfs are identical except for a shift of $2\Delta G^f(\infty)$ in $z$. Examples of the pdfs in the Gaussian case are shown in the schematic figure 2.

## 2.4   Jarzynski equality

We denote the averaging with respect to the forward process by:

$$\langle X(w) \rangle_f = \int_{-\infty}^{\infty} X(w)\mathrm{pdf}^f(w)dw \tag{30}$$

The Jarzynski equality is then expressed by:

$$\langle e^{-\beta w} \rangle_f = e^{-\beta \Delta G} \tag{31}$$

In the Gaussian case we have:

$$\langle e^{-\beta w} \rangle_f = \frac{1}{\sqrt{2\pi}\sigma_W} \int_{-\infty}^{\infty} e^{-\beta w} \exp(-\frac{(w - W^f)^2}{2\sigma_W^2})dw \tag{32}$$

and verifying the Jarzynski equality reduces to equating the powers of the exponential:

$$-(w - W^f)^2 - 2\beta \sigma_W^2(w - \Delta G) = -(w - W_0)^2. \tag{33}$$

The only non-trivial solution is:

$$\beta = \frac{2Q}{\sigma_w^2} \quad \text{and} \quad W_0 = W^r. \tag{34}$$

The Jarzynski equality applies when $\beta$ is a constant, independent of the forcing protocol $\mathbf{F}$ and $T$. At first sight eq. (34) is

astonishing, as the Jarzynski equality expressed by eq. (31) seems to be a statement on the free energy which solely depends on the integrated mode. Equation (34), however, connects the heat dissipated in the shear mode to the work fluctuations, which





are only due to the shear mode. Furthermore, we have seen that the dynamics of the two modes are unrelated. This apparent inconsistency is resolved by multiplying eq. (31) by $e^{\beta \Delta G}$ on both sides. It is then apparent that an average of the exponential of $\Delta G - w$ is taken, which only depends on the shear mode. Note that when thermal fluctuations are considered $\beta^{-1} = k_B T$ and more generally for the Ornstein-Uhlenbeck process $\beta_D^{-1} = \frac{R}{SM}$, which relates the fluctuations to the dissipation and shows the connection of the Jarzynski equality to the fluctuation dissipation relation and the fluctuation dissipation theorem (see Wirth, 2019, 2018).

Experiments can also be performed for different values of $\beta_D$ (see the schematic fig. 2). If the turbulence level decreases, $\beta_D$ increases and the dynamics converges towards a deterministic process. Note that neither $\Delta G - W$ nor $\sigma_w^2$ does depend on $u_I(0)$, in the case of vanishing Coriolis force this is equivalent to Galilean invariance.

Furthermore, neither the work nor the free energy depend on the relaxation process and in an experiment it is not necessary to wait for the relaxation to the stationary state to obtain the free energy. It is only necessary to repeat the experiment sufficiently many times to obtain a statistically significant results and use Jarzynski equality to obtain the free energy. The work does, however, depend on $\mathbf{u}_S(0)$ and so we have to start from equilibrium ($\langle \mathbf{u}_S(0) \rangle = 0$ and $\langle \mathbf{u}_S(0)^2 \rangle = \beta_D^{-1}$). The Jarzynski equality also shows that, as $\sigma_w^2 > 0$, there have to be (rare) paths for which the work performed is smaller than the free energy. This is easily seen as $e^{-x} < 1$ for $x > 0$. In thermodynamics these paths are sometimes referred to as "violations of the second law of thermodynamics". However, due to the convexity of the exponential function $\langle e^x \rangle \geq e^{\langle x \rangle}$ (called Jensen's inequality) and therefore $\langle w \rangle \geq \Delta G$ and the second law of thermodynamics is verified in an average sense, it is a statistical law.

## 2.5 Crooks relation

The Jarzynski equality (JE) considers an average with respect to the forward process, whereas the Crooks relation (CR) compares the pdf's of the forward and reverse process, without any averaging, it states:

$$\frac{\mathrm{pdf}^f(w)}{\mathrm{pdf}^r(-w)} = \exp(\beta_D[w - \Delta G]) = \exp(-\beta_D q). \tag{35}$$

where $q = \Delta G - w$ is the negative dissipation along a single trajectory with work $w$ and $Q = \langle q \rangle$ by definition. The CR is also useful to determine $\Delta G$, it is the value $w$ where the graphs of the forward pdf and the reverse pdf of the negative argument cross, where $\mathrm{pdf}^f(w) = \mathrm{pdf}^r(-w)$ (see fig. 2). When the shape of the forward and reverse pdf agree, the free energy can be also obtained via: $\mathrm{pdf}^r(w) = \mathrm{pdf}^f(w + 2\Delta G)$. The first method is useful when $\beta_D$ is small and the second when it is large. The CR considers the pdf and the JE which is concerned with averages can be derived from it through dividing eq. (35) by $\exp(\beta_D q)$ multiplying by $\mathrm{pdf}^r(-w)$ and integrating over $w$ from $-\infty$ to $\infty$. In cyclic or stationary processes the free energy gain is vanishing and the revers pdf equals the forward pdf and the CR simplifies to the detailed fluctuation theorem.

In the Gaussian case described above the CR is obtained by a straight forward calculation:

$$\frac{\mathrm{pdf}^f(w)}{\mathrm{pdf}^r(-w)} = \exp\left(\frac{1}{2\sigma_w^2}(-w^2 + 2W^f w - (W^f)^2 + w^2 + 2W^r w + (W^r)^2)\right) = \exp(\beta_D[w - \Delta G]) = \exp(-\beta_D q). \tag{36}$$





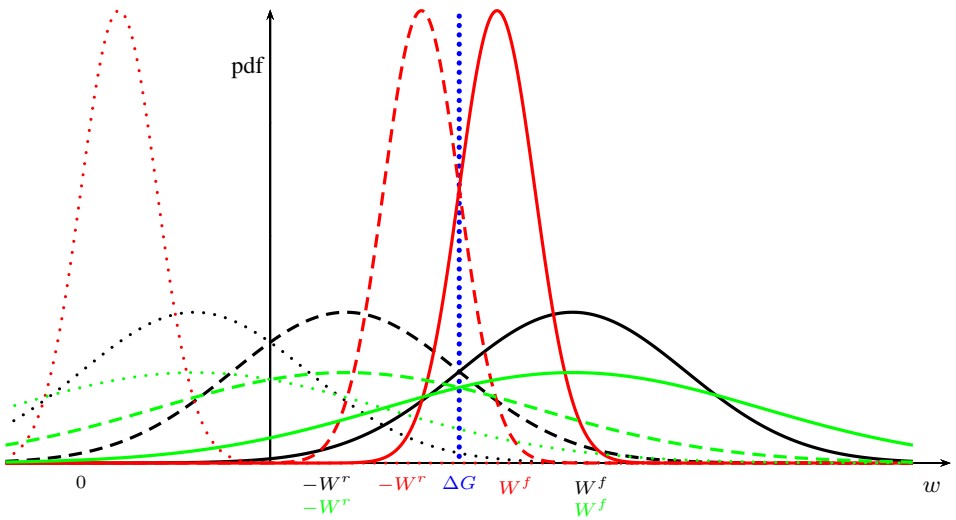

**Figure 2.** The pdfs of the forward (pdf$^f(w)$, full lines), the backward (pdf$^r(w)$, dotted lines) and the negative of the backward (pdf$^r(-w)$, dashed lines) processes are shown. The averages $W^f$ and $-W^r$ are given on the horizontal axis for the processes. The red-line represents an experiment that was performed at a slower rate, less work is provided on average and fluctuations are smaller as compared to the experiment corresponding to the black line. The green-line represents an experiment that was performed at a lower dynamic-$\beta_D$ as compared to the experiment corresponding to the black line, averages agree but fluctuations are higher. The dotted-lines are shifted by $2\Delta G$ to the left with respect to the full lines of the same color. The full-lines and the dashed lines of the same color all intersect at $w = \Delta G$ and the dotted-lines and the dashed-lines of the same color all intersect at $w = 0$. For an experiment performed at $\beta_D = \infty$ (zero temperature) or an experiment performed at infinitely slow rate $T \to \infty$ all the pdfs are the delta function $\delta(w - \Delta G)$ (blue line) and the dynamics is deterministic.

## 2.6 Integral Fluctuation Theorem

Note also that when the CR holds:

$$1 = \left\langle \frac{\mathrm{pdf}^r(-w)}{\mathrm{pdf}^f(w)} \right\rangle_f = \langle \exp(\beta_D q) \rangle_f. \tag{37}$$

This is the integral fluctuation theorem, it shows that there exists trajectories with $q > 0$, "violations of the second law of thermodynamics". It is proven by using Jensen's inequality: $1 = \langle \exp(\beta_D q) \rangle \geq \exp(\beta_D Q)$, which leads to $Q \geq 0$. The integral fluctuation theorem is a reformulation of the JE in terms of dissipated heat.

In cyclic or stationary processes the free energy gain is vanishing. When a force is applied, it typically drives the system, does work, which is dissipated to heat. Rare events when the work is negative and heat does work must exist, following the above derived results.



## 3 Results

The work relations are investigated for a hierarchy of models of air-sea interaction. This not only favors a pedagogical discussion of the subject but also helps to emphasize critical points in the application of the theory exposed above. The simpler models, which are given by eqs. (9), are called 0D-models as the variables have no spatial dependence. The friction force between the two layers is parameterized by linear Rayleigh friction, which allows for analytical solutions. In these linear models which are subject to Gaussian noise (through the $\zeta_x$ and $\zeta_y$ terms in (9)) the pdfs of the work are Gaussian random variables, which are determined by their mean, their variance and their temporal correlation. In this case the work theorems are algebraic relations between the means and the variances which can be calculated analytically using stochastic calculus. The first model of interest, referred to as OD1C model, does not include the Coriolis force ($f = 0$) and the dynamics in the two horizontal directions (the two components of the velocity vector) are independent. The analytical solution for this model is given in section 3.1. The 0D1C model represents the simplest example in which work theorems can be discussed and solved analytically by employing Newton's laws and solving stochastic differential equations. It can also be shown that in this case the work theorems are a consequence of Galilean invariance. When the Coriolis force is added the 0D2C model (9) is recovered and the dynamics in the two horizontal directions interact. The Coriolis force also adds several conceptual difficulties to the problem. First, for Brownian motion of particles subject to a Coriolis force, detailed balance is lost as the dynamics is not time reversible. Second, Galilean invariance is broken, even for the deterministic part of the problem. Third, the application of a reverse protocol depends on the timing. The same force can increase or reduce the free energy depending on when, at which phase, it is applied. We calculate the work theorem analytically by employing Newton's laws and solving stochastic differential equations in section 3.2.

We then discuss in section 3.3 a nonlinear model with a vertical dependence in the atmosphere and ocean (1D2C-model), that describes the non-linear interaction of the two planetary boundary layers. It is described by eqs. (1), (2), (3) and (5). The model is deterministic except for the drag coefficient, which has a stochastic part. All results concerning this model are obtained through numerical integration of the corresponding governing equations.

### 3.1 The linear 0D1C-model

The solution of the 0D1C model introduced in section 2.1 is:

$$u_a(t) = u_a(0) - (u_a(0) - u_o(0))\frac{m}{M}\left(1 - e^{-MSt}\right) + \int_0^t \frac{1}{M}\left(1 + m\,e^{-MS(t-t')}\right)\mathcal{F}(t')\,dt' + \frac{m}{M}\int_0^t e^{-MS(t-t')}\zeta(t')dt' \quad \text{(38a)}$$

$$u_o(t) = u_o(0) + (u_a(0) - u_o(0))\frac{1}{M}\left(1 - e^{-MSt}\right) + \int_0^t \frac{1}{M}\left(1 - e^{-MS(t-t')}\right)\mathcal{F}(t')\,dt' - \frac{1}{M}\int_0^t e^{-MS(t-t')}\zeta(t')dt' \quad \text{(38b)}$$





where $\mathcal{F}(t)$ is the deterministic forcing of the synoptic atmosphere and $\zeta$ a random force. The steps to obtain such solution are given in section A1. In terms of the integrated mode and the shear mode the equivalent solution is:

$$u_I(t) = \int_0^t \mathcal{F}(t')dt + u_I(0) \tag{39a}$$

$$u_S(t) = \int_0^t e^{-SM(t-t')}[\mathcal{F}(t') + \zeta(t')]dt' + u_S(0)e^{-SMt}, \tag{39b}$$

In the following we consider that the noise $\zeta(t)$ is delta correlated in time:

$$\langle \zeta(t)\zeta(t')\rangle = 2R\delta(t-t'). \tag{40}$$

with $R$ a positive scalar.

### 3.1.1 Constant Forcing

The simplest case is a constant force $\mathcal{F}(t)$ of amplitude $\mathcal{F}_0$ during the interval $I = [0, T]$, such forcing satisfies the symmetry
required for a reverse protocol as given by eq. (22). Note that in a linear model the results obtained with such forcing are general because every forcing can be approximated by a sum of step-function forcings, or an integral of infinitesimal step functions. The dynamics of a sum of step functions or an integral, is the sum or integral of the dynamics of the individual forcings. The solution for $0 \leq t \leq T$ is:

$$u_I(t) = \mathcal{F}_0 t + u_I(0) \tag{41}$$

$$u_S(t) = \frac{\mathcal{F}_0}{SM}(1 - e^{-SMt}) + u_S(0)e^{-SMt} \tag{42}$$

and for $t \geq T$

$$u_I(t) = u_I(T) \tag{43}$$

$$u_S(t) = u_S(T)e^{-SM(t-T)} = \frac{\mathcal{F}_0}{SM}(e^{-SM(t-T)} - e^{-SMt}) + u_S(0)e^{-SMt} \tag{44}$$

The work can be separated in the work done on the shear mode $W_S$ and on the integrated mode $W_I$. The work the system
absorbs as well as, how much is absorbed by each mode depends on the state of the system and on the initial condition. The work is:

$$
\begin{aligned}
W &= \int_0^T \mathcal{F}u_a dt = W_S + W_I = \int_0^T \frac{\mathcal{F}}{M}(mu_S + u_I)dt \\
&= \frac{\mathcal{F}_0^2}{2M}T^2 + \frac{\mathcal{F}_0}{M}(u_I(0) + \frac{m\mathcal{F}_0}{SM})T + \frac{m\mathcal{F}_0}{SM^2}(u_S(0) - \frac{\mathcal{F}_0}{SM})(1 - e^{-SMT}) \tag{45} \\
W_I &= \frac{\mathcal{F}_0^2}{2M}T^2 + \frac{\mathcal{F}_0}{M}u_I(0) \tag{46} \\
\end{aligned}
$$

$$W_S = \frac{m\mathcal{F}_0^2}{S^2M^3}(SMT - 1 + e^{-SMT}) + \frac{m\mathcal{F}_0}{SM^2}(1 - e^{-SMT})u_S(0) \tag{47}$$





The kinetic energy:

$$V(T) \;=\; \frac{1}{2}(u_a^2 + mu_o^2) = \frac{1}{2M}(u_I^2 + mu_S^2) = \frac{1}{2M}\left((\mathcal{F}_0 T + u_I(0))^2 + m(\frac{\mathcal{F}_0}{SM}(1 - e^{-SMT}) + u_S(0)e^{-SMT})^2\right) \tag{48}$$

$$V(t) \;=\; \frac{1}{2M}\left((\mathcal{F}_0 T + u_I(0))^2 + m(\frac{\mathcal{F}_0}{SM}(e^{-SM(t-T)} - e^{-SMt}) + u_S(0)e^{-SMt})^2\right) \tag{49}$$

for $t > T$; if $t \in I$ replace $T$ by $t$ in the above equation.

The energy difference is:

$$\Delta V(T) \;=\; V(T) - V(0)$$
$$=\; \frac{1}{2M}(u_I^2 + mu_S^2) = \frac{1}{2M}\left((\mathcal{F}_0 T + u_I(0))^2 - u_I(0)^2 + m(\frac{\mathcal{F}_0}{SM} + (u_S(0) - \frac{\mathcal{F}_0}{SM})e^{-SMT})^2 - mu_S(0)^2\right)$$
$$=\; \frac{\mathcal{F}_0^2}{2M}T^2 + \frac{\mathcal{F}_0 u_I(0)}{M}T + \frac{m}{2M}(\frac{\mathcal{F}_0}{SM})^2$$
$$+\; \frac{m\mathcal{F}_0}{SM^2}(u_S(0) - \frac{\mathcal{F}_0}{SM})e^{-SMT} + \frac{m}{2M}(u_S(0) - \frac{\mathcal{F}_0}{SM})^2 e^{-2SMT} - \frac{m}{2M}u_S(0)^2 \tag{50}$$

$$\Delta V(\infty) \;=\; \frac{\mathcal{F}_0^2 T^2}{2M} + \frac{\mathcal{F}_0 T u_I(0)}{M} - \frac{m}{2M}u_S(0)^2 \tag{51}$$

The free energy is the energy difference in the integrated mode, as the shear mode relaxes to zero in equilibrium, when the forcing has resided. Therefore it equals the work performed on the integrated mode $W_I$.

$$\Delta G(T) \;=\; \frac{\mathcal{F}_0^2 T^2}{2M} + \frac{\mathcal{F}_0 T u_I(0)}{M} = W_I \tag{52}$$

The free energy only varies when work is performed on the system. When the process is infinitely slow the free energy equals
the work. The energy dissipated is:

$$Q(0,T) \;=\; SM\frac{m}{M}\int_0^T u_S^2 dt' = Sm\int_0^T (\frac{\mathcal{F}_0}{SM}(1 - e^{-SMt'}) + u_S(0)e^{-SMt'})^2 dt'$$
$$=\; Sm((\frac{\mathcal{F}_0}{SM})^2 T + \frac{2\mathcal{F}_0}{(SM)^2}(\frac{\mathcal{F}_0}{SM} - u_S(0))(e^{-SMT} - 1) - \frac{1}{2SM}(\frac{\mathcal{F}_0}{SM} - u_S(0))^2(e^{-2SMT} - 1))$$
$$=\; \frac{m\mathcal{F}_0^2}{SM^2}T - \frac{2m\mathcal{F}_0}{SM^2}(u_S(0) - \frac{\mathcal{F}_0}{SM})(e^{-SMT} - 1) - \frac{m}{2M}(u_S(0) - \frac{\mathcal{F}_0}{SM})^2(e^{-2SMT} - 1)$$
$$=\; \frac{m\mathcal{F}_0^2}{SM^2}T - \frac{2m\mathcal{F}_0}{SM^2}(u_S(0) - \frac{\mathcal{F}_0}{SM})e^{-SMT} - \frac{m}{2M}(u_S(0) - \frac{\mathcal{F}_0}{SM})^2 e^{-2SMT} + \frac{mu_S(0)^2}{2M} - \frac{3m\mathcal{F}_0^2}{2S^2 M^3} + \frac{m\mathcal{F}_0 u_S(0)}{SM^2} \tag{53}$$

$$Q(T,t) \;=\; Sm\int_T^t u_S(T)^2 e^{-2SMt'} dt' = \frac{mu_S(T)^2}{2M}(1 - e^{-2SMt}) \tag{54}$$

### 3.1.2   Forward and Reverse Process (deterministic)

The free energy starting from rest is:

$$\Delta G \;=\; \frac{\mathcal{F}_0^2 T^2}{2M}. \tag{55}$$





The forward process starts from rest and is forced with amplitude $\mathcal{F}_0$ for a period $T$ and is then let to relax:

$$u_I(0) = u_S(0) = 0 \qquad \rightarrow \qquad u_I(\infty) = \mathcal{F}_0, \ u_S(\infty) = 0 \tag{56}$$

$$\Delta G^f(\infty) = \Delta G, \tag{57}$$

$$W^f = \frac{\mathcal{F}_0^2}{2M} T^2 + \frac{m}{S}\left(\frac{\mathcal{F}_0}{M}\right)^2 T - m\frac{\mathcal{F}_0^2}{S^2 M^3}(1 - e^{-SMT}) = \Delta G + \frac{m\mathcal{F}_0^2 T^2}{2M} A(T) = \Delta G(1 + mA(T)) \tag{58}$$

$$A(T) = 2\frac{e^{-SMT} - 1 + SMT}{(SMT)^2} \tag{59}$$

$$Q^f(0,\infty) = m\Delta G A(T) \tag{60}$$

The reverse process starts from the converged state is forced with amplitude $-\mathcal{F}_0$ for a period $T$ and then relaxes to rest:

$$u_I(-\infty) = u_S(-\infty) = 0 \qquad \leftarrow \qquad u_I(\infty) = \mathcal{F}_0, \ u_S(\infty) = 0 \tag{61}$$

$$\Delta G^r(-\infty) = -\Delta G, \tag{62}$$

$$W^r = -\frac{\mathcal{F}_0^2}{2M} T^2 + \frac{m}{S}\left(\frac{\mathcal{F}_0}{2M}\right)^2 T - m\frac{\mathcal{F}_0^2}{S^2 M^3}(1 - e^{-SMT}) = -\Delta G + \frac{m\mathcal{F}_0^2 T^2}{M} A(T) = \Delta G(-1 + mA(T)) \tag{63}$$

$$Q^r(0,\infty) = Q^f(0,\infty) = m\Delta G A(T) \tag{64}$$

Note that $-W^r \leq \Delta G \leq W^f$ which is a statement of the second law of thermodynamics. When the process is reversible then the equalities apply. It is interesting to note that during a very slow process ($T \to \infty$ while keeping $\mathcal{F}_0 T$ fixed), the process approaches the reversible limit. Furthermore $2\Delta G = W^f - W^r$.

### 3.1.3 Forward and Reverse Process (stochastic)

When noise is added in the linear model it does not interfere with the deterministic dynamics but just adds to it. Furthermore, the force is prescribed (therefore deterministic) and the randomness in the work provides solely from the fluctuations in $u_a$ and as randomness resides only in the shear-mode the fluctuations in the work $w' = \int \mathcal{F}_0 u'_S dt$ come from fluctuations of the shear mode $u'_S$. The work values are Gaussian variables with a mean that is the value of the deterministic part ($\langle w^f \rangle = W^f$, $\langle w^r \rangle = W^r$) and the variance is given by the variance of the Ornstein-Uhlenbeck process integrated over the time interval $T$ (see appendix A2):

$$\sigma_W^2(T) = \frac{m\mathcal{F}_0^2 T^2}{M} \langle(\overline{u_S'}^T)^2\rangle = \frac{2Rm\mathcal{F}_0^2 T^2}{M(SM)^3 T^2}(\exp(-SMT) - 1 + SMT) = \frac{Rm}{SM}\Delta G A(T) \tag{65}$$

where $A(T)$ is given in eq. (59), showing a relation between the difference of the work to the free energy (the dissipated energy) and the stochastic fluctuations, this is the fluctuation dissipation theorem (see Wirth, 2019). In this case $-W^r \leq \Delta G^f(\infty) \leq W^f$ is true on average only, individual trajectories can be exceptions. Note that:

$$\sigma_W^2(0) = \frac{Rm\Delta G}{SM} \tag{66}$$

$$\lim_{T \to \infty} \sigma_W^2(T) = \frac{Rm\Delta G}{SM}\frac{2}{SMT}, \tag{67}$$

the instantaneous correlation is recovered when the averaging time vanishes and the $T^{-1}$ law for averaging times larger than the correlation time. The pdf's are:

$$\text{pdf}^f(w) = \frac{1}{\sqrt{2\pi}\sigma_W}\exp(-\frac{(w-W^f)^2}{2\sigma_W^2}) \tag{68}$$

$$\text{pdf}^r(w) = \frac{1}{\sqrt{2\pi}\sigma_W}\exp(-\frac{(w-W^r)^2}{2\sigma_W^2}) \tag{69}$$

### 3.1.4 Jarzynski equality and Crooks relation

In order to apply the JE to the present problem we identify the heat by $Q = W_S$ and the free energy by $\Delta G = W_I$ and eq. (34) leads to

$$\beta_D = \frac{SM}{R}. \tag{70}$$

This proves that the JE applies with the standard dynamic-$\beta_D$ of the Ornstein-Uhlenbeck process.

Note, that in the above all dependence is on the product $\mathcal{F}_0 T$ and not on the factors independently in this linear problem. Experiments can also be performed at different temperature Also, Galilean invariance is assured as neither $\Delta G - W$ nor $\sigma_w^2$ does depend on $u_I(0)$. Furthermore, neither the work nor the free energy depend on the relaxation process, so the above is always true and in an experiment it is not necessary to wait for the relaxation to the stationary state to obtain the free energy. It is only necessary to do the experiment in a sufficiently many times and use JE to obtain the free energy. The work does, however depend on $u_S(0)$ and so we have to start from equilibrium ($u_S(0) = 0$). As discussed in section 2.4, JE also shows that there has to be (rare) paths for which the work performed is smaller than the free energy, but $\langle w \rangle \geq \Delta G$ and the second law of thermodynamics is verified in an average sense, it is a statistical law. Expressed in terms of the dissipation along a trajectory, the JR leads to $\langle e^{-\beta_D q} \rangle = 1$ and again $\langle q \rangle \geq 0$ on average, but paths exist with negative dissipation.

The CR is obtained by a straight forward calculation introducing $W^f = W_I + W_S$ and $W^r = -W_I + W_S$:

$$\frac{\text{pdf}^f(w)}{\text{pdf}^r(-w)} = \exp\left(\frac{1}{2\sigma_w^2}(-w^2 + 2W^f w - (W^f)^2 + w^2 + 2W^r w + (W^r)^2)\right) = \exp(\beta_D[w - \Delta G]) = \exp(\beta_D \Delta Q). \tag{71}$$

### 3.2 The linear 0D2C-model

The calculations performed for the one-component model will now be extended to the two component model where the two components interact through the Coriolis force (see appendix B). The solutions of the integrated mode $\mathbf{u}_I(t)$ and the shear $\mathbf{u}_S(t)$ mode are given by eqs. (B10a) and (B10b). From these equations it follows that the work is:

$$W = W_I + W_S = \frac{1}{M}\int_0^T \mathcal{F}(t)\mathcal{C}_I(\mathcal{F})(t)dt + \frac{m}{M}\int_0^T \mathcal{F}(t)\mathcal{C}_S(\mathcal{F})(t)dt \tag{72}$$

where $\mathcal{C}_I$ and $\mathcal{C}_S$ are defined in (B9). The free energy is again $\Delta G = W_I$. The free evolving system typically relaxes to a state where the integrated mode performs undamped inertial oscillations, which is non stationary. When a forcing is applied, the work and free energy change depends on the phase of the integrated mode.





### 3.2.1 Constant forcing

We start from a system at rest and apply the force constant $\mathcal{F}$ of amplitude $\mathcal{F}_0$ for a time interval $T$ to the $x$-component.

$$W_I = \frac{\mathcal{F}_0^2}{Mf^2}(1-\cos(fT)) \tag{73}$$

$$W_S = \frac{m\mathcal{F}_0^2}{M((SM)^2+f^2)}\left(SMT + \frac{1}{(SM)^2+f^2}\left[((SM)^2-f^2)\cos(fT)-2SMf\sin(fT))e^{-SMT}-(SM)^2-f^2\right]\right) \tag{74}$$

For $f=0$, this is equivalent to eqs. (46) and (47).

As the model is linear all statistics are Gaussian and the statistical properties are completely described by the first order moments, which are described by the deterministic equations and the second order moments. Assuming the noise to be isotropic in the horizontal ($\langle\zeta_x^2\rangle = \langle\zeta_y^2\rangle = 2R$, $\langle\zeta_x\zeta_y\rangle = 0$) we obtain for the random part:

$$\mathbf{u}_S' = \begin{pmatrix} m(\mathcal{C}_S(\zeta_x)-\mathcal{S}_S(\zeta_y)) \\ m(\mathcal{S}_S(\zeta_x)+\mathcal{C}_S(\zeta_y)) \\ -\mathcal{C}_S(\zeta_x)+\mathcal{S}_S(\zeta_y) \\ -\mathcal{S}_S(\zeta_x)-\mathcal{C}_S(\zeta_y) \end{pmatrix} \tag{75}$$

where again the $\mathcal{C}_S$ and $\mathcal{S}_S$ are given in (B9).

### 3.2.2 Jarzynski equality and Crooks relation

Note, that for the work fluctuations only the $x$-component, to which the forcing applies has to be considered, that is the random fluctuations in $u' = u_a'$ averaged over the interval $T$:

$$\overline{u'}^T(t) = \frac{me^{-SMt}}{T}\int_0^T e^{-SMT'}\int_0^{t+T'} e^{SMt'}\left[\zeta_u(t')\cos(f(t+T'-t'))+\zeta_v(t')\sin(f(t+T'-t'))\right]dt'dT' \tag{76}$$

Due to the linearity the process is Gaussian with a vanishing mean-value and a variance (see appendix B2):

$$\langle[\overline{u'}^T(t)]^2\rangle = \frac{2R}{Sm(\mathcal{F}_0T)^2}W_S \tag{77}$$

We obtain:

$$\langle[\overline{w'}^T(t)]^2\rangle = \frac{m(\mathcal{F}_0T)^2}{M}\langle[\overline{u'}^T(t)]^2\rangle = \frac{2R}{SM}W_S \tag{78}$$

which leads to:

$$\beta_D = \frac{SM}{R}, \tag{79}$$

which is the same dynamic-$\beta_D$ than in the one dimensional non-rotating case.

### 3.3 The one-dimensional non-linear boundary-layer model

In this model we resolve part of the dynamics in the interior of the atmospheric and the oceanic layer explicitly. The model consists of eq. (1) and the boundary conditions (2) and (3). The thickness of the atmospheric layer is $h_a = 300\text{m}$ and for the oceanic





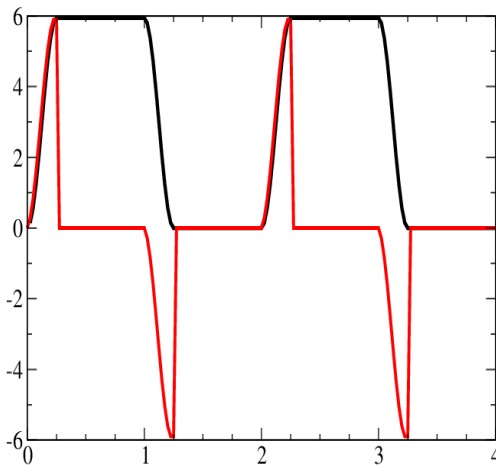

**Figure 3.** Evolution of the free energy (black) and the work performed on the integrated mode (red) from the numerical integration is shown. The evolution is deterministic and periodic and results agree with the analytic solution.

layer it is $h_o = 30$m, with densities $\rho_a = 1\,\text{kg m}^{-3}$ and $\rho_o = 1000\,\text{kg m}^{-3}$. The Coriolis parameter is $f = 10^{-4}\,\text{s}^{-1}$. The verti-
cal viscosity in the atmosphere is calculated by a Turbulent Kinetic Energy (TKE) scheme with a shear-based length scale (see
Sec. 4.1 in Lemarié et al., 2020) and in the ocean a K-Profile Parameterization (KPP, see Sec. 2c in McWilliams and Huckle,
2006) is used. The shear between the layers is calculated using eq. (5). The randomness is introduced through the friction coef-
ficient $c_d$ it is given by the square of a random Gaussian variable with a variance $c_d^m = 1.2 \cdot 10^{-3}$ and an exponential correlation
in time with a decay time of $t_{cd}^{\text{exp.1}} = f^{-1}$ in experiment 1 and $t_{cd}^{\text{exp.2}} = 10 f^{-1}$ in experiment 2. This is justified by the fact that
friction coefficient depends on a variety of physical properties as the wave spectrum and velocity of propagation, as well as the
stratification and boundary-layer turbulence in the atmosphere and the ocean, which all vary in space and time. This typically
leads to a large variability of the measure $c_d$ coefficient (see e.g. Csanady, 2001; Oost et al., 2002; Large, 2006; Patton et al.,
2019). Results from two sets of numerical experiments, exp1 and exp2, are presented here. The structure of the model is again
the same as shown in fig. 1, the random part is given by $T$ (red color in the figure), all other interactions are presented through
deterministic equations.

For this model the free energy is still given by the kinetic energy of the integrated mode, as all other motion decays when
forcing resides. It is governed by the same equation than in the linear 0D Coriolis model, that is, its dynamics is independent
of the shear and the internal modes in the atmosphere and the ocean. We call $T = 4\pi f^{-1} = 1$day. The forcing protocol is a
constant force that is applied in the intervals $[jT, (2+.25)T]$ for the days $j = 1, ..., n$. The forcing is applied to the $x$-component
only through a large large-scale pressure gradient via a geostrophic velocity: $\mathcal{F}_x = -(-1)^j f v_G$ and $\mathcal{F}_y = 0$ in eqs. (1a) and





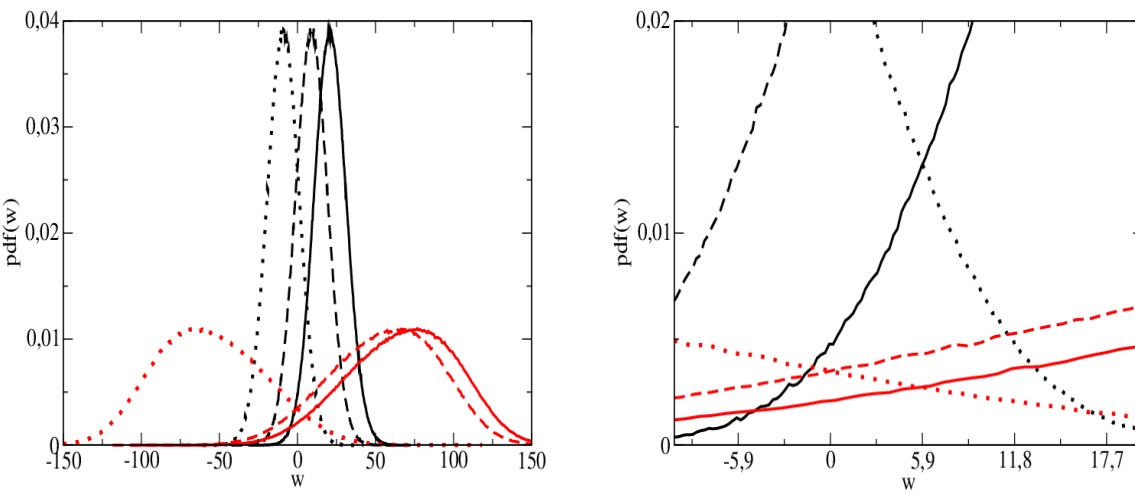

**Figure 4.** The forward pdf (pdf$^f(z)$; full-line), the backward pdf (pdf$^r(z)$; dashed-line)), and (pdf$^r(-z)$; dotted-line), exp1 is in black and exp2 in red. Full and dotted lines intersect at $\Delta z = 5.9$ as can be seen in the zoom (left). Exp2 is clearly non Gaussian.

(1b) respectively, that is, the forward and reverse forcing alternate periodically. The periodic work applied to the integrated mode and the evolution of the free energy are shown in fig. 3, both agree with the analytic solution.

The dynamics of the shear mode is governed by the same equations than in the linear 0D Coriolis model with a deterministic forcing and a friction at the air sea interface. The difference to the 0D model is that the dynamics of the internal modes within the atmosphere and the ocean are explicitly resolved and they influence the shear force that acts on the shear mode. That is, the 5 stochastic term in the 0D models mimics the influence of the internal modes in the atmosphere and the ocean. The 1D model also resolves the shear modes, not only between the atmosphere and the ocean, but also within them. These modes interact in a non-linear way and exchange energy, which is ultimately dissipated when the external forcing resides. In the 1D model the internal modes within the atmosphere and within the ocean interact through the surface friction term and the internal eddy 10 viscosities. In more involved 2D or 3D models, not studied here, they also interact through non-linear horizontal advection.

The numerical model to solve the above discussed equations, is a variation of the one used in (Lemarié et al., 2020). There are 20 levels in the atmosphere and 20 in the ocean, with first grid points at $\delta_a = 5$m and $\delta_o = 1$m, in the atmosphere and the ocean, respectively. The time step of the integration is $10\pi$s and. For both experiments, the integration consists of a spin-up of $4 \cdot 10^3$days followed by an integration of $4.4 \cdot 10^3$days the ensemble size is of each integration is $10^3$ and 10 integrations 15 where performed. The total ensemble size, for each experiment, is therefore $2.2 \cdot 10^7$, when we suppose ergodicity (note that the protocol repeats every 2 days).

The work performed on the atmosphere is now a random process. The numerical results show that the average work performed on the atmosphere in the forward process in the two experiments is $\Delta W^{\text{exp.1}} = 21.0$Jm$^{-2}$ and $\Delta W^{\text{exp.2}} = 67.3$Jm$^{-2}$,



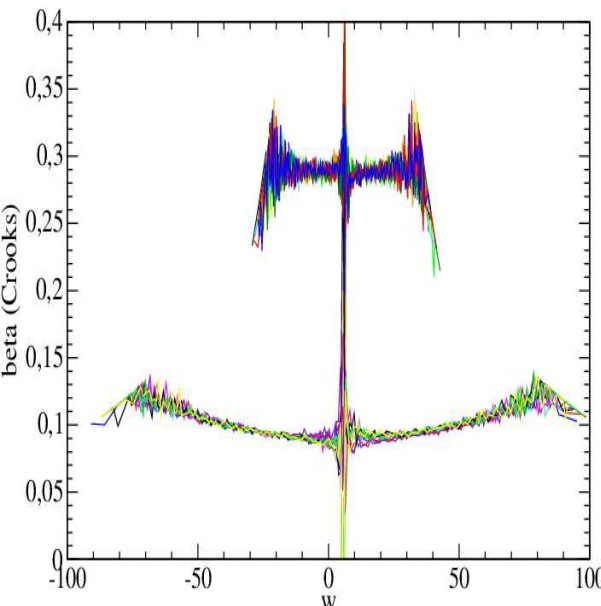

**Figure 5.** Fig. gives $\beta_{CR}$ as a function of the work $w$ calculated with the CR (eq. 80, for exp.1 (upper graph) and exp.2 (lower graph). For each experiment ten statistically independent realizations are superposed.

while only a small part of this work drives the integrated mode, contributes to the free energy $\Delta G = 6/101 \approx 5.94$. Its value can be calculated analytically, it is independent on the friction process and therefore equal in both experiments. Results of the numerical integration are shown in fig. 4 where the different pdfs are visualized. The standard deviations of the pdfs are $\sigma^{\mathrm{exp.1}} = 10.3 \mathrm{Jm}^{-2}$ and $\sigma^{\mathrm{exp.2}} = 34.8 \mathrm{Jm}^{-2}$ They are close to but significantly different from Gaussian with a skewness (third

standardized moment) of $\mu_3^{\mathrm{exp.1}} = 0.03$ and $\mu_3^{\mathrm{exp.2}} = -0.20$ . In this case the verification of the work theorems no-longer reduces to algebraic relations between the first and second-order moments, but the whole shape of the pdfs has to be considered. The forward pdf and the backward pdf flipped at $z = 0$ (pdf$^r(-z)$) intersect at $\Delta G$, in both experiments (fig. 4), within statistical error as predicted by the CR. The forward pdf and the backward pdf shifted by $2\Delta G$ superpose within statistical error, as can be seen in fig. 4. This is a consequence of the independence of the deterministic dynamics of the integrated mode, on the

rest of the dynamics and the symmetry of the forcing protocol given in eq.(23). The same figure shows clearly that probability for a forward event with work smaller than the free energy $\Delta G$ is non negligible, the equivalent of such events in thermal processes are related to as "violations of the second law of thermodynamics". Note that the probability of a forward event with negative work are also present.

   We numerically found the JE $\langle \exp(-\beta_{JE}(w - \Delta G)) \rangle_f = 1$ in the two experiments to be satisfied for $\beta_{JE}^{\mathrm{exp.1}} = 0.115$ and

15 $\beta_{JE}^{\mathrm{exp.2}} = 0.290$, respectively. Where we denote by $\beta_{JE}$ the value of $\beta_D$ obtained from the data through the JE.



For evaluating the CR we plotted:

$$\beta_{CR} = \frac{1}{w - \Delta G} \ln \left( \frac{\mathrm{pdf}^f(w)}{\mathrm{pdf}^r(-w)} \right). \tag{80}$$

Where we denote by $\beta_{CR}$ the value of $\beta_D$ obtained from the data through the CR. Note that near $\Delta G$ this expression is strongly dependent on the bin size, where the nominator and denominator go to zero, which makes a numerical evaluation difficult and

leads to strong oscillations. We clearly see that $\beta_{CR}^{\mathrm{exp.2}}$ is close to but significantly different from a constant and that $\beta_{JE}^{\mathrm{exp.2}}$ is a good approximation for values around the maximum of $\mathrm{pdf}^f(\mathrm{w})$. The dimension of $\beta$ is the inverse of an energy and the obvious question is to see how it can be related to the dynamics. In the Gaussian case eq. (70) shows that it is equal to the ratio of the heat dissipated over one cycle and the variance of the work:

$$\beta_{\mathrm{Gauss}} = \frac{(W_f + W_r)}{\sigma^2}, \tag{81}$$

For the present non Gaussian model these calculations lead to $\beta_{\mathrm{Gauss}}^{\mathrm{exp.1}} = 0.29$ and $\beta_{\mathrm{Gauss}}^{\mathrm{exp.2}} = 0.10$, it is equal to $\beta_{CR}$ and $\beta_{JE}$ for the exp1 (with a close to Gaussian pdf) and close to $\beta_{CR}$ and $\beta_{JE}$ for exp2.

## 4   Discussion

We started by introducing the concept of work theorems in a simple model of air-sea interaction. In this case the JE and the CR can be obtained analytically. We then performed the same calculations on a model including a Coriolis force. In that case

the time reversibility is broken and the dynamics lags detailed balance, which is at the basis of the original proofs of the JE and the CR in Hamiltonian system. Analytical integrations of the stochastic differential equations governing the dynamics of the system proof the existence of the JE and the CR. It furthermore shows that the limit of $f \to 0$ is well defined and the non-rotating solution is obtained

In the applications of work theorems where fluctuations arise from thermal dynamics, the thermodynamic-$\beta$ is fixed by the

temperature of the heat bath. In the system considered here there is no external heat bath, but the fluctuations are generated by the external forcing and the internal dynamics. The different value of the dynamic-$\beta$ in the two experiments comes, therefore, at no surprise, as the fluctuations now arise from the dynamics of the shear mode and the internal modes in the atmosphere and the ocean, which clearly differ between both experiments. In terms of heat fluctuations this means that the system is not thermostated, there is no outside heat-bath that keeps the temperature (or $\beta$) constant. In exp.2 the variation of the drag

coefficient is ten-times slower than in exp.1 and the dynamics of the shear mode and the internal modes in the atmosphere and the ocean have more time to adjust to its instantaneous value. The drag not only influences the work but also the variability leading to a dynamic-$\beta$ that depends on $w$. The result that a variation of the dynamic-$\beta$ is undetectable in exp.1 and small in exp.2, allows to define an average $\beta$ in the present dynamics. This shows the pertinence of the work theorems by Jarzynski and Crooks in the present context as they not only apply to exp.1 but also to exp.2 in which the fluctuations are slower than the

forcing protocol. This is important as forcing protocols and turbulence levels vary over a large continuum of time scales.





The physical interpretation of the dynamic-$\beta$ or its inverse, often called effective temperature (Feitosa and Menon (2004)) or characteristic energy (Ciliberto et al. (2004)), is given by eq. (81) as the ratio of the heat dissipated over one cycle and the variance of the work.

## 5   Conclusions

We have shown that the modern concepts of non-equilibrium statistical mechanics can be applied to large scale environmental fluid dynamics, where fluctuations are not thermal but come from the turbulent fluid motion. We have demonstrated that the concepts of dynamic-beta, that is the equivalent of "temperature" in dynamical systems, can be extended to air-sea interaction using the formalism developed by Jarzynski and Crooks. It is important to note that work theorems are valid for forces of arbitrary amplitude, they are not a perturbative theory. It is, to the best of our knowledge, the first time that the concepts of

work relations are investigated in geophysics and climate science. We successfully adapted the work theorems to the subject of air-sea momentum transfer but it can, in the same way, be applied to other components of the climate system.

Work theorems have also important practical applications. When the work pdfs of the forward and backward process are obtained, the free energy of the system and the dissipated energy can be obtained and a mechanical efficiency of the air-sea momentum transfer calculated. This is key in understanding the energetics of the climate system. When the CR applies, the

likeliness of some rare and extreme events can be obtained from parts of the pdf that represent likely events. Furthermore, when work theorems are found to apply in observations they represent an important tool to evaluate numerical integrations and parameterizations in models of the environmental dynamics.

The mechanics of air-sea momentum transfer has advanced considerably since the pioneering work of Ekman (1905) and is today an active field of research, Duhaut and Straub (2006); McWilliams and Huckle (2006); Zhai et al. (2012); Shrira et al.

(2020). In an environment fluctuating on a vast continuum of scales in space-and-time, the statistical mechanics has to be advanced.

The difficulty in performing simulations in air-sea interaction is the large difference in the characteristic time scales of the fast atmosphere and the slow ocean, the stiffness of the problem. Therefore integrations of the fast atmospheric dynamics are necessary with a long spin-up, as the ocean has to be in a statistically stationary state followed by a long integration to obtain

a statistical significance ensemble of ocean states. When observations are considered, the stiffness asks for observations over extended periods of time which are just becoming available.

Similar problems appear when the interaction of other components of the climate system are considered. Air-sea interaction is just one example where work relations between fluctuating components of the climate system increase our understanding. Their extension to other components is straightforward.





## Appendix A: The linear 0D1C-model

### A1 Solution

We consider a state vector given by the atmospheric and oceanic velocity:

$$\mathbf{u}(t) = \begin{pmatrix} u_a(t) \\ u_o(t) \end{pmatrix}.$$
(A1)

The evolution equation for the 0D1C model (i.e. model (9) with $f = 0$) can be written in a matrix form as

$$\partial_t \mathbf{u} = \mathbf{P}\mathbf{u} + \mathbf{F}^{\text{ext}},$$
(A2)

with:

$$\mathbf{P} = -S \begin{pmatrix} m & -m \\ -1 & 1 \end{pmatrix}, \quad \text{and} \quad \mathbf{F}^{\text{ext}} = \mathcal{F} \begin{pmatrix} 1 \\ 0 \end{pmatrix} + \frac{\zeta}{M} \begin{pmatrix} m \\ -1 \end{pmatrix}$$
(A3)

where $\mathcal{F}$ is the deterministic forcing of the synoptic atmosphere on the atmospheric boundary layer and $\zeta$ the random noise

parameterizing internal turbulent motion which does not act on the integrated momentum.

The first step to solve the system of ODEs (A2) is to diagonalize $\mathbf{P}$. The eigenvalues $\lambda_j$ and associated eigenvectors $\mathbf{e}_j$ of $\mathbf{P}$ are:

$$\lambda_1 = 0, \ \mathbf{e}_1 = \begin{pmatrix} 1 \\ 1 \end{pmatrix}; \quad \lambda_2 = -SM, \ \mathbf{e}_2 = \begin{pmatrix} m \\ -1 \end{pmatrix}$$
(A4)

with $M = m + 1$. The square matrix $\mathbf{P}$ is thus diagonalizable and can be decomposed as

$$\mathbf{P} = \mathbf{A}\mathbf{D}\mathbf{A}^{-1}, \quad \text{with} \quad \mathbf{A} = \begin{pmatrix} 1 & m \\ 1 & -1 \end{pmatrix}, \quad \mathbf{D} = \begin{pmatrix} 0 & 0 \\ 0 & -SM \end{pmatrix}.$$
(A5)

Furthermore, $M\mathbf{A}^{-1} = \mathbf{A}$ and also

$$\mathbf{A}^{-1}\mathbf{u} = \frac{1}{M}(\mathbf{A}\mathbf{u}) = \frac{1}{M} \begin{pmatrix} u_a + mu_o \\ u_a - u_o \end{pmatrix} = \frac{1}{M} \begin{pmatrix} u_I \\ u_S \end{pmatrix}$$

which shows that the integrated $u_I$ and shear $u_S$ modes defined in (10) are eigenmodes of the dynamics. We can thus re-express equation (A2) with the unknown $\mathbf{u}$ as

$$\partial_t \mathbf{u}_M = \mathbf{D}\mathbf{u}_M + \mathbf{A}\mathbf{F}^{\text{ext}}, \quad \mathbf{u}_M = \begin{pmatrix} u_I \\ u_S \end{pmatrix}$$
(A6)

with the unknown $\mathbf{u}_M$. Because $\mathbf{D}$ is a diagonal matrix the two ODEs in (A6) are decoupled and can be solved separately. As reported in (39), we easily find that

$$\mathbf{u}_M(t) = \begin{pmatrix} 1 & 0 \\ 0 & e^{-MSt} \end{pmatrix} \mathbf{u}_M(0) + \int_0^t \begin{pmatrix} \mathcal{F}(t') \\ e^{-MS(t-t')}[\mathcal{F}(t') + \zeta(t')] \end{pmatrix} dt'$$
(A7)





and the solution in terms of the original unknowns $u_a$ and $u_o$ given in (38) is simply obtained using the relation $\mathbf{u} = (\mathbf{A}\mathbf{u}_M)/M$ to get:

$$\mathbf{u}(t) = \frac{1}{M} \left\{ \begin{pmatrix} 1 + m\,e^{-MSt} & m(1 - e^{-MSt}) \\ 1 - e^{-MSt} & m + e^{-MSt} \end{pmatrix} \mathbf{u}(0) + \int_0^t \begin{pmatrix} \mathcal{F}(t') + m\,e^{-MS(t-t')} \left[\mathcal{F}(t') + \zeta(t')\right] \\ \mathcal{F}(t') - \quad e^{-MS(t-t')} \left[\mathcal{F}(t') + \zeta(t')\right] \end{pmatrix} dt' \right\}$$
(A8)

which can be recast as:

$$u_a(t) = u_a(0) - (u_a(0) - u_o(0))\frac{m}{M}\left(1 - e^{-MSt}\right) + \int_0^t \frac{1}{M}\left(1 + m\,e^{-MS(t-t')}\right)\mathcal{F}(t')\,dt' + \frac{m}{M}\int_0^t e^{-MS(t-t')}\zeta(t')dt' \quad \text{(A9a)}$$

$$u_o(t) = u_o(0) + (u_a(0) - u_o(0))\frac{1}{M}\left(1 - e^{-MSt}\right) + \int_0^t \frac{1}{M}\left(1 - e^{-MS(t-t')}\right)\mathcal{F}(t')\,dt' - \frac{1}{M}\int_0^t e^{-MS(t-t')}\zeta(t')dt' \quad \text{(A9b)}$$

## A2 Variance

The deterministic and the stochastic dynamics are statistically independent so that when calculating statistical moments we can ignore the deterministic one (i.e. $\mathcal{F}(t)$ will be ignored). In the following we note $u'_S$ the random part of $u_S$. The solution of the shear mode is given by eq. (39b). Its variance is, using the fact that $\langle \zeta(t)\zeta(t') \rangle = 2R\delta(t - t')$ (see eq. (40)) and that $\langle \zeta(t) \rangle = 0$:

$$
\begin{aligned}
\langle u'_S(t)^2 \rangle &= \left\langle \left( \int_0^t e^{-SM(t-t')}\zeta(t')dt' + u_S(0)e^{-SMt} \right)^2 \right\rangle \\
&= \int_0^t \int_0^t e^{-SM(t-t')}e^{-SM(t-t'')}\langle \zeta(t'')\zeta(t') \rangle dt''dt' + e^{-2SMt}\langle u'_S(0)^2 \rangle \\
&= 2R\int_0^t e^{-SM(t-2t')}dt' + e^{-2SMt}\langle u_S(0)^2 \rangle = \frac{R}{SM}(1 - e^{-2SMt}) + e^{-2SMt}\langle u'_S(0)^2 \rangle.
\end{aligned}
$$
(A10)

It is important to note that there are two different averages involved in the above equation, all denoted by the same symbol $\langle \cdot \rangle$. One is over the noise and the other over the initial conditions. Using the same symbol is justified as the initial conditions are due to the same statistical noise applied prior to $t = 0$. In a statistically stationary process the variance is independent of the time $t$ and therefore

$$\langle u'_S(t)^2 \rangle = \langle u'_S(0)^2 \rangle = \frac{R}{SM}$$
(A11)

Such consistency condition is extensively used throughout this manuscript.

For the work theorems the focus is on the statistics of averages over a time interval $T$. Using eq. (A7) we obtain:

$$
\begin{aligned}
\overline{u'_S}^T(t) &= \frac{1}{T}\int_0^T u'_S(t + T')dT' = \frac{1}{T}\int_0^T \int_0^{t+T'} e^{-SM(t+T'-t')}\zeta(t')dt' + u_S(0)e^{-SM(t+T')}dT' \\
&= \frac{e^{-SMt}}{T}\int_0^T \int_0^{t+T'} e^{-SM(T'-t')}\zeta(t')dt'dT' + \frac{e^{-SMt}}{SMT}(1 - e^{-SMT})u_S(0)
\end{aligned}
$$
(A12)





Its variance is:

$$
\begin{aligned}
\langle [\overline{u_S'}^T(t)]^2 \rangle &= \frac{e^{-2SMt}}{T^2} \int_0^T \int_0^T \int_0^{t+T'} \int_0^{t+T''} e^{-SM(T'-t'+T''-t'')} \langle \zeta(t')\zeta(t'') \rangle dt''dt'dT''dT' + \frac{e^{-2SMt}}{(SMT)^2}(1-e^{-SMT})^2 \langle u_S(0)^2 \rangle \\
&\qquad [\text{define}: \quad \tilde{T} = min(T',T'')] \\
&= \frac{2Re^{-2SMt}}{T^2} \int_0^T \int_0^T \int_0^{t+\tilde{T}} e^{-SM(T'+T''-2t')}dt'dT''dT' + \frac{e^{-2SMt}}{(SMT)^2}(1-e^{-SMT})^2 \frac{R}{SM} \\
&= \frac{R}{SMT^2} \int_0^T \int_0^T e^{-SM(T'+T''-2\tilde{T})} - e^{-SM(T'+T''+2t)}dT''dT' + \frac{e^{-2SMt}}{(SMT)^2}(1-e^{-SMT})^2 \frac{R}{SM} \\
&= \frac{2R}{SMT^2} \int_0^T \int_0^{T'} e^{-SM(T'-T'')} - e^{-SM(T'+T''+2t)}dT''dT' + \frac{e^{-2SMt}}{(SMT)^2}(1-e^{-SMT})^2 \frac{R}{SM} \\
&= \frac{2R}{(SMT)^2} \int_0^T 1 - e^{-SMT'} + e^{-SM(2T'+2t)} - e^{-SM(T'+2t)}dT' + \frac{e^{-2SMt}}{(SMT)^2}(1-e^{-SMT})^2 \frac{R}{SM} \\
&= \frac{2R}{(SMT)^2}[T + \frac{e^{-SMT}-1}{SM} - e^{-2SMt}(\frac{e^{-2SMT}-1-2e^{-SMT}+2}{2SM})] + \frac{e^{-2SMt}}{(SMT)^2}(1-e^{-SMT})^2 \frac{R}{SM} \\
&= \frac{2R}{(SM)^3T^2}[SMT + e^{-SMT} - 1] = \frac{2R}{Sm(\mathcal{F}_0T)^2}W_S. \qquad (A13)
\end{aligned}
$$

Note that for $T \ll SM$ we have $\langle [\overline{u_S'}^T(t)]^2 \rangle = \langle u_S(t)^2 \rangle$ and for $T \gg SM$ we obtain $\langle [\overline{u_S'}^T(t)]^2 \rangle = 2\langle u_S(t)^2 \rangle/(SMT)$.

## Appendix B: The linear 0D2C-model

The calculations performed for the one-component model in previous section will now be extended to the two component model where the two components interact through the Coriolis force.

### B1 Solution

To simplify the algebra we temporarily manipulate complex quantities in this subsection. For the 0D2C model given in (9) we consider a state vector given by

$$
\mathbf{y}(t) = \begin{pmatrix} U_a(t) \\ U_o(t) \end{pmatrix}, \qquad (B1)
$$

where $U_a$ and $U_o$ are complex quantities such that $U_a = u_a + iv_a$ and $U_o = u_o + iv_o$. The general evolution equation satisfied by $\mathbf{y}$ is

$$
\partial_t \mathbf{y} = \mathbf{P}_f \mathbf{y} + \mathbf{F}_f^{\text{ext}}, \qquad (B2)
$$





with:

$$\mathbf{P}_f = \begin{pmatrix} -if - Sm & Sm \\ S & -if - S \end{pmatrix} \quad \text{and} \quad \mathbf{F}_f^{\text{ext}} = \begin{pmatrix} \mathcal{F} \\ 0 \end{pmatrix} + \frac{1}{M}(\zeta_x + i\zeta_y)\begin{pmatrix} m \\ -1 \end{pmatrix} \tag{B3}$$

where $F_u$, $\zeta_u$ and $\zeta_v$ are real-valued functions. The matrix $\mathbf{P}_f$ is diagonalizable and can be decomposed as

$$\mathbf{P}_f = \mathbf{A}\mathbf{D}_f\mathbf{A}^{-1}, \quad \text{with} \quad \mathbf{A} = \begin{pmatrix} 1 & m \\ 1 & -1 \end{pmatrix}, \quad \mathbf{D}_f = \begin{pmatrix} -if & 0 \\ 0 & -if - SM \end{pmatrix}. \tag{B4}$$

We recall that $M\mathbf{A}^{-1} = \mathbf{A}$ and introducing the complex numbers $U_I = u_I + iv_I$ and $U_S = u_S + iv_S$ corresponding to the integrated and shear modes we have

$$\mathbf{A}^{-1}\mathbf{y} = \frac{1}{M}(\mathbf{A}\mathbf{y}) = \frac{1}{M}\mathbf{y}_M, \quad \mathbf{y}_M = \begin{pmatrix} U_I \\ U_S \end{pmatrix}.$$

Re-expressing the original system of ODEs in terms of $\mathbf{y}_M$ we obtain

$$\partial_t \mathbf{y}_M = \mathbf{D}_f \mathbf{y}_M + \mathbf{A}\mathbf{F}_f^{\text{ext}}$$

and we obtain two independent ODEs for the complex functions $U_I(t)$ and $U_S(t)$

$$\partial_t U_I = -if U_I + \mathcal{F}(t) \tag{B5a}$$

$$\partial_t U_S = (-if - SM)U_S + \mathcal{F}(t) + (\zeta_x(t) + i\zeta_y(t)) \tag{B5b}$$

whose solutions are

$$U_I(t) = U_I(0)e^{-ift} + \int_0^t e^{-if(t-t')}\mathcal{F}(t')dt' \tag{B6a}$$

$$U_S(t) = U_S(0)e^{-(if+SM)t} + \int_0^t e^{-(if+SM)(t-t')}\left[\mathcal{F}(t') + (\zeta_x(t') + i\zeta_y(t'))\right]dt' \tag{B6b}$$

Taking the real and imaginary parts of $U_I(t)$ we obtain

$$u_I(t) = \cos(ft)u_I(0) + \sin(ft)v_I(0) + \int_0^t \cos(f(t-t'))\mathcal{F}(t')dt' \tag{B7a}$$

$$v_I(t) = \cos(ft)v_I(0) - \sin(ft)u_I(0) + \int_0^t \sin(f(t-t'))\mathcal{F}(t')dt' \tag{B7b}$$

Now considering the shear mode we have

$$u_S(t) = e^{-SMt}\left(\cos(ft)u_S(0) + \sin(ft)v_S(0)\right) + \int_0^t e^{-SM(t-t')}\left[(\mathcal{F}(t') + \zeta_x(t'))\cos(f(t-t')) + \zeta_y(t')\sin(f(t-t'))\right]dt'$$

$$v_S(t) = e^{-SMt}\left(\cos(ft)v_S(0) - \sin(ft)u_S(0)\right) + \int_0^t e^{-SM(t-t')}\left[\zeta_y(t')\cos(f(t-t')) - (\mathcal{F}(t') + \zeta_x(t'))\sin(f(t-t'))\right]dt'$$





Note that for $f = 0$ we easily recover the solutions from the 0D1C model. To further simplify those solutions we introduce the following notations

$$
\begin{aligned}
\mathcal{C}_I(x) &= \int_0^t \cos(f(t-t'))x(t')dt' & \mathcal{C}_S(x) &= \int_0^t e^{-SM(t-t')}\cos(f(t-t'))x(t')dt' \\
\mathcal{S}_I(x) &= \int_0^t \sin(f(t-t'))x(t')dt' & \mathcal{S}_S(x) &= \int_0^t e^{-SM(t-t')}\sin(f(t-t'))x(t')dt'
\end{aligned}
\tag{B9}
$$

to reformulate the solutions of the 0D2C model as

$$
\qquad \mathbf{u}_I(t) = \begin{pmatrix} \cos(ft) & \sin(ft) \\ -\sin(ft) & \cos(ft) \end{pmatrix} \mathbf{u}_I(0) + \begin{pmatrix} \mathcal{C}_I(\mathcal{F}) \\ \mathcal{S}_I(\mathcal{F}) \end{pmatrix}
\tag{B10a}
$$

$$
\mathbf{u}_S(t) = \begin{pmatrix} \cos(ft) & \sin(ft) \\ -\sin(ft) & \cos(ft) \end{pmatrix} e^{-SMt}\mathbf{u}_S(0) + \begin{pmatrix} \mathcal{C}_S(\mathcal{F}) + \mathcal{C}_S(\zeta_x) + \mathcal{S}_S(\zeta_y) \\ \mathcal{C}_S(\zeta_y) - \mathcal{S}_S(\mathcal{F}) - \mathcal{S}_S(\zeta_x) \end{pmatrix}
\tag{B10b}
$$

**B2  Variance**

For the sake of clarity we use the notations $c = \cos(ft)$ and $s = \sin(ft)$ in the following. We consider a Gaussian process with a variance:

$$
\begin{aligned}
\qquad \langle [u_S'(t)]^2 \rangle &= 2R\int_0^t e^{-2SM(t-t')}dt' + e^{-2SMt}(c^2\langle u_S(0)^2\rangle + s^2\langle v_S(0)^2\rangle + 2cs\langle u_S(0)v_S(0)\rangle) \\
&= \frac{R}{SM}(1 - e^{-2SMt}) + e^{-2SMt}\langle u_S(0)^2\rangle = \frac{R}{SM}.
\end{aligned}
\tag{B11}
$$

It the noise is turned off at $t = 0$ we have the deterministic evolution (using: $\cos(t+\alpha)\cos(t+\beta) + \sin(t+\alpha)\sin(t+\beta) = \cos\alpha\cos\beta + \sin\alpha\sin\beta$):

$$
\overline{u_S^0}^T(t) = \frac{e^{-SMt}}{T}\int_0^T e^{-SMT'}(\cos f(t+T')u_S(0) + \sin f(t+T')v_S(0))dT'
\tag{B12}
$$

and its variance is given by:

$$
\begin{aligned}
\langle [\overline{u_S^0}^T(t)]^2 \rangle &= \frac{e^{-2SMt}}{T^2}\int_0^T\int_0^T e^{-SM(T'+T'')}\langle [\cos f(t+T')u_S(0) + \sin f(t+T')v_S(0)] \\
&\qquad [\cos f(t+T'')u_S(0) + \sin f(t+T'')v_S(0)]\rangle dT''dT' \\
&= \frac{Re^{-2SMt}}{SMT^2}\int_0^T\int_0^T e^{-SM(T'+T'')}(\cos fT'\cos fT'' + \sin fT'\sin fT'')dT''dT' \\
&= \frac{Re^{-2SMt}}{SMT^2}\left[ (\int_0^T e^{-SMT'}\cos fT'dT')^2 + (\int_0^T e^{-SMT'}\sin fT'dT')^2 \right] \\
&= \frac{Re^{-2SMt}(e^{-2SMT} - 2e^{-2SMT}\cos(fT) + 1)}{SMT^2((SM)^2 + f^2)}.
\end{aligned}
\tag{B13}
$$





It cancels the time dependent part of the variance due to the noise (see eq. (B14)), making the total variance time independent.

This is another expression of the consistency condition mentioned in the previous section. The total variance is:

$$
\begin{aligned}
\langle [\overline{u'_S}^T(t)]^2 \rangle \;=\; & \langle \left[ \overline{\mathcal{C}_S(\zeta_u) + \mathcal{S}_S(\zeta_v)}^T + \overline{u^0_S}^T(t) \right]^2 \rangle \\[2mm]
=\; & \frac{e^{-2SMt}}{T^2} \int_0^T \int_0^T e^{-SM(T'+T'')} \int_0^{t+T'} \int_0^{t+T''} e^{SM(t'+t'')} \\[2mm]
& \langle [\zeta_u(t')\cos(f(t+T'-t')) + \zeta_v(t')\sin(f(t+T'-t'))] [\zeta_u(t'')\cos(f(t+T''-t'')) + \zeta_v(t'')\sin(f(t+T''-t''))] \rangle \\[2mm]
& dt''\, dt'\, dT''\, dT' + \langle [\overline{u^0_S}^T(t)]^2 \rangle \\[2mm]
& [\text{define}: \quad \tilde{T} = min(T',T'')] \\[2mm]
=\; & \frac{2Re^{-2SMt}}{T^2} \int_0^T \int_0^T e^{-SM(T'+T'')} \int_0^{t+\tilde{T}} e^{2SMt'} \cos(fT')\cos(fT'') + \sin(fT')\sin(fT'') dt'\, dT''\, dT' + \langle [\overline{u^0_S}^T(t)]^2 \rangle \\[2mm]
=\; & \frac{2Re^{-2SMt}}{T^2} \int_0^T \int_0^T e^{-SM(T'+T'')} \int_0^{t+\tilde{T}} e^{2SMt'} \cos(f(T'-T'')) dt'\, dT''\, dT' + \langle [\overline{u^0_S}^T(t)]^2 \rangle \\[2mm]
=\; & \frac{R}{SMT^2} \int_0^T \int_0^T (e^{SM(2\tilde{T}-T'-T'')} - e^{SM(-2t-T'-T'')}) \cos(f(T'-T'')) dT''\, dT' + \langle [\overline{u^0_S}^T(t)]^2 \rangle \\[2mm]
=\; & \frac{2R}{SMT^2} \int_0^T \int_0^{T'} e^{-SM(T'-T'')} \cos(f(T'-T'')) dT''\, dT' = \frac{2R}{SMT^2} \int_0^T \mathcal{C}_S(1)dt = \frac{2R}{Sm(\mathcal{F}_0 T)^2} W_S. \qquad (\text{B14})
\end{aligned}
$$

Where $W_S$ is given by eq. (74). It is important to note that the last equality is equal to the last equality in eq. (A13) and that in the limit $f \to 0$ the solutions of the non-rotating case are obtained in all the formulas.





*Acknowledgements.* This work was funded by the French LEFE (Les Enveloppes Fluides et l'Environnement) MANU (méthodes MAthé-matiques et NUmériques) program through project FASIL.





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
