# Peer review of "Jarzynski equality and Crooks relation for local models of air-sea interaction"

_Earth System Dynamics, 2020_

## Referee Comment (RC1) · Anonymous Referee #1 · 11 Jan 2021

Authors analyze the applicability of Jarzynski equality and Crooks relation, (taken from non-equilibrium statistical mechanics) in local air-sea models, subjected both to Coriolis force and a time-varying external forcing protocol. Authors show the quite ample range of situations in which those relations apply. Authors present their results in a sequence of models of increasing complexity, first analytical, then numerical models, in a rigorous and pedagogical form.

The manuscript is suitable for publication in ESD after the correction of some minor remarks.

Pg 2, line 31. When speaking about betaD, explain better its meaning and how it can be obtained. In the present context betaD is the inverse a temperature of a turbulent kinetic energy. Include a statement here explaining that. What should be the equivalent

to entropy in the present context?

Pg 5, line 2. The external forcing in the ocean equations vanishes. Why? By hypothesis? Explain.

Pg 8 Note in the text that, since the Coriolis force is orthogonal to the speed vector, it does not interfere neither in the work nor in the heat.

Pg. 9 line 1, BetaG is the variation of free energy: G(B)-G(A). In order to be consistent, should the free energy be defined as G(A)=G(A)-G(infinity) ? The state infinity is ambiguous and must be avoided or carefully defined. Clarify please. Solve other cases where free energy or variation of free energy is misused (e.g. pg. 15, line 22).

Pg 9, Eqs. 25,27 say which are the dependent variables of Q (e.g. Q(0,infinity) ).

Pg. 11 line 4. The parameter R must be defined here. Normally, according to Wirth (2019), the parameter R measures the strength of the delta-correlated fluctuating force, taken as a white noise process. Include that expression here (See eq. 40).

Pg. 14 In the rhs of Eq. 46, the second term must be multiplied by T

Pg. 16, Eqs. 56 and 61, $u_I (\infty)=F_0 T$ since $F_0$ is an acceleration. Please verify the consistency of physical dimensions.

Pg. 19, line 16. The upper bound of the time interval can be smaller than the lower bound for j=3,…. Please check.

Pg. 21, lines 11-12. Authors say 'probability for a forward event with work smaller than the free energy ïĄĎG is non negligible, the equivalent of such events in thermal processes are related to as "violations of the second law of thermodynamics".' Could you give a number for that probability.

Fig. 5 is never referred in the text. Include its discussion.

[Figure]

2020.

---

## Referee Comment (RC2) · Anonymous Referee #2 · 15 Feb 2021

The manuscript addresses the applicability of the Jarzynski equality and the Crooks relations from non-equilibrium statistical mechanics to study the exchanges of momentum between the ocean and the atmosphere. A hierarchy of local models is employed. Two more idealized models are solved analytically, which include a time-varying (white-noise) external forcing with and without Coriolis. Finally, a more elaborate model considering the interaction between atmospheric and oceanic boundary layer is integrated numerically. This is a relevant contribution, employing modern concepts of non-equilibrium statistical mechanics to deepen the understanding of large-scale geophysical fluid flows. The manuscript is well-structured, the methods are results are clearly described, despite the complexity of the problem. In my opinion, the manuscript is acceptable for publication after minor changes.

[Figure]

- The study focuses on the air-sea exchange of momentum. This should be clearly stated in the abstract and conclusions, since air-sea interactions encompass a much broader range of processes.

- The discussion of the Figures must be improved in the text. The authors should better guide the reader through shown the results, namely for Fig. 3, but also for Fig. 2. Moreover, Figure 5 is not mentioned in the text. The authors should either discuss this figure or remove it

- Page 23 line 14: "This is key in understanding the energetics of the climate system". I believe that providing the a few sentences explaining the implications for the energetics of the climate system and the future directions of research would be important and could broaden the interest of the present manuscript to a wider audience.

- Page 1 line 1 "We show using a hierarchy of local models of air-sea interaction that the most prominent of the work theorems, the Jarzynski equality and the Crooks relation can be applied to air-sea interaction". This provides an important summary of the manuscript. Maybe improve the clarity of the sentence by re-phrasing to something like "We show that the Jarzynski equality and the Crooks relation can be applied to air-sea interaction using. . ."

- Page 22 Line 13: "We started by introducing the concept of work theorems in a simple model of air-sea interaction" – Provide a few words describing (summarizing) this simple model.

- The notation when referencing figures in the text should be consistent throughout the text (Fig. our figure, e.g. Page 10 line 12 and Page 20 line 2 )

---

## Author Comment (AC1) · 8 Mar 2021

[esd, manuscript]copernicus

picture

amsmath

Answers to both reviewers:

Dear Reviewers,

We are grateful to both reviewers for their corrections and comments as they have increased the quality of the paper. Please find our detailed answers and corrections to

both reviewers comments (reproduced in black) below, written in blue. The corrections performed to the manuscript are given in red and an updated version of the manuscript with the corrections highlighted in red is provided.

Sincerely,

Achim Wirth and Florian Lemarié

Reviewer 1:

Authors analyze the applicability of Jarzynski equality and Crooks relation, (taken from non-equilibrium statistical mechanics) in local air-sea models, subjected both to Coriolis force and a time-varying external forcing protocol. Authors show the quite ampler range of situations in which those relations apply. Authors present their results in a sequence of models of increasing complexity, first analytical, then numerical models, in a rigorous and pedagogical form.

The manuscript is suitable for publication in ESD after the correction of some minor remarks.

Pg 2, line 31. When speaking about betaD, explain better its meaning and how it can be obtained. In the present context betaD is the inverse a temperature of a turbulent kinetic energy. Include a statement here explaining that. What should be the equivalent to entropy in the present context?

I now changed the sentence to:

It is the inverse of a "temperature" that is in the present context, of a turbulent kinetic energy.

Putting temperature in quotes, as we are not sure that most of the readers of ESD are familiar with dynamic temperature and I am afraid that using to many technical terms will hinder the acceptance of the here presented concepts by the community of earth-

system scientists. For the same reason we do not talk about entropy. It is noted in the original paper by **?**, that "This establishes $W \geq \Delta F$ directly from a microscopic, Hamiltonian basis, rather than by invoking the increase of entropy." And we find it appealing that the work relations can be discussed without evoking the concept of entropy and we do like-wise. The concepts are of course related but discussing the relation does, to our understanding, only lead to confusion and hides the fact that the work theorems can be obtained independent of the increase of entropy. Furthermore in the non-Gaussian behavior of the numerical model several choices of entropy are possible, further complicating the discussion.

Pg 5, line 2. The external forcing in the ocean equations vanishes. Why? By hypothesis? Explain.

Yes, we now added:

The upper ocean is mainly forced at its surface by the wind shear. Forcing due to the dynamics of the deeper or surrounding ocean are not considered by the model.

Pg 8 Note in the text that, since the Coriolis force is orthogonal to the speed vector, it does not interfere neither in the work nor in the heat.

Yes, we now added at the end of section 2.2:

It is important to note that the Coriolis parameter does not explicitly appear in the equation of the work or the heat as the Coriolis force is orthogonal to the local velocity. However, the Coriolis parameter strongly influences the dynamics, that is $u_a$ and $u_o$. Through this influence, it has a determining role on the work and heat budget.

Pg. 9 line 1, BetaG is the variation of free energy: G(B)-G(A). In order to be consistent, should the free energy be defined as G(A)=G(A)-G(infinity) ? The state infinity is ambiguous and must be avoided or carefully defined. Clarify please. Solve other cases where free energy or variation of free energy is misused (e.g. pg. 15, line 22).

We now added:

The energy in the integrated mode changes only when a forcing is applied, so it varies only during the protocol $A \to B$, or its inverse, is applied, whereas the internal energy $V$ varies before and after.

Pg 9, Eqs. 25,27 say which are the dependent variables of Q (e.g. Q(0,infinity)).

To emphasize the dependence on the protocol we now changed the sentence before the eqs. to:

During the forward process, starting from rest and applying the protocol $\tilde{F}_{A \to B}(t)$, we have:

Pg. 11 line 4. The parameter R must be defined here. Normally, according to Wirth(2019), the parameter R measures the strength of the delta-correlated fluctuating force, taken as a white noise process. Include that expression here (See eq. 40).

We now added:

, with an auto-correlation of the delta-correlated fluctuating force characterized by the variable $R$ (defined below through eq. (40)),

Pg. 14 In the rhs of Eq. 46, the second term must be multiplied by T

Oups, thank you ! now corrected.

Pg. 16, Eqs. 56 and 61, $u_I(\infty) = F_0 T$ since $F_0$ is an acceleration. Please verify the consistency of physical dimensions.

Done.

Pg. 19, line 16. The upper bound of the time interval can be smaller than the lower bound for j=3,.... Please check.

Done.

Pg. 21, lines 11-12. Authors say 'probability for a forward event with work smaller than the free energy $\Delta G$ is non negligible, the equivalent of such events in thermal

processes are related to as "violations of the second law of thermodynamics".' Could you give a number for that probability.

In the Gaussian case there is of course a analytic expression. We now added:

The probability of such a violation to occur in the Gaussian case can be expressed using the error function as: $\mathrm{erf}((\Delta G - W^f)/\sigma_W))$

Thank you for the hint !

Fig. 5 is never referred in the text. Include its discussion.

The figure is now referred twice.

Reviewer 2:

The manuscript addresses the applicability of the Jarzynski equality and the Crooks relations from non-equilibrium statistical mechanics to study the exchanges of momentum between the ocean and the atmosphere. A hierarchy of local models is employed. Two more idealized models are solved analytically, which include a time-varying (white-noise) external forcing with and without Coriolis. Finally, a more elaborate model considering the interaction between atmospheric and oceanic boundary layer is integrated numerically. This is a relevant contribution, employing modern concepts of non-equilibrium statistical mechanics to deepen the understanding of large-scale geophysical fluid flows. The manuscript is well-structured, the methods are results are clearly described, despite the complexity of the problem. In my opinion, the manuscript is acceptable for publication after minor changes.

- The study focuses on the air-sea exchange of momentum. This should be clearly stated in the abstract and conclusions, since air-sea interactions encompass a much broader range of processes.

The referee is right, the paper clearly focuses on the exchange of momentum, but the formalism developed can be applied to other exchanges as e.g. heat, as also in this case the there is a strong (even stronger) discrepancy in heat capacity per volume between the atmosphere and the ocean. The origin is in all cases the strong density difference between air and water

The first sentence in the abstract is now changed to:

We show that the most prominent of the work theorems, the Jarzynski equality and the Crooks relation, can be applied to the momentum transfer at the air-sea interface using a hierarchy of local models.

In the conclusion section we changed:

"We have demonstrated that the concepts of dynamic-beta, that is the equivalent of "temperature" in dynamical systems, can be extended to the momentum transfer at the air-sea interface using the formalism developed by Jarzynski and Crooks."

and:

"The momentum transfert at the air-sea interface is just one example where work relations between fluctuating components of the climate system increase our understanding."

- The discussion of the Figures must be improved in the text. The authors should better guide the reader through shown the results, namely for Fig. 3, but also for Fig.2. Moreover, Figure 5 is not mentioned in the text. The authors should either discuss this figure or remove it

Fig. 2 is an illustration of the theory of work theorems discussed in the text. Its caption is long and self-explanatory. We now added in the text:

Examples of the pdfs in the Gaussian case for the forward and reverse processes, as well as the pdfs the reverse process flipped at the origin are shown for different

values of the in the schematic Fig. **??** for illustration.

Concerning the discussion of Fig. 3 we now added:

and the periodic response to the periodic forcing is clearly visible. This verifies that the dynamics of integrated mode is not affected by the random fluctuations of the shear coefficient.

- Page 23 line 14: "This is key in understanding the energetics of the climate system". We believe that providing the a few sentences explaining the implications for the energetics of the climate system and the future directions of research would be important and could broaden the interest of the present manuscript to a wider audience.

  We are convinced that work theorems can be applied to the interaction of other components of the climate system but feel uneasy to make suggestions into fields that we have not explored. For a broader view we now added:

  For a discussion of the ocean circulation kinetic energy we refer the reader to ? and for a spatio-temporal variability of the momentum transfer to the ocean to ?.

  In this paper we present a detailed example of how work theorems can be applied to momentum transfer at the air-sea interface, which can guide the applications, to interacting components with fluctuations.

- Page 1 line 1 "We show using a hierarchy of local models of air-sea interaction that the most prominent of the work theorems, the Jarzynski equality and the Crooks relation can be applied to air-sea interaction". This provides an important summary of the manuscript. Maybe improve the clarity of the sentence by rephrasing to something like "We show that the Jarzynski equality and the Crooks relation can be applied to air-sea interaction using..."

  We changed the sentence to:

We show that the most prominent of the work theorems, the Jarzynski equality and the Crooks relation, can be applied to the momentum transfer at the air-sea interface using a hierarchy of local models.

- Page 22 Line 13: "We started by introducing the concept of work theorems in a simple model of air-sea interaction" – Provide a few words describing (summarizing) this simple model.

  We now added:

  in which the atmosphere and ocean were represented by their corresponding mixed layer.

- The notation when referencing figures in the text should be consistent throughout the text (Fig. our figure, e.g. Page 10 line 12 and Page 20 line 2 )

  We now use Fig. X and figure, when no number follows, everywhere.